



**Impacts of ice-nucleating particles from marine aerosols on mixed-phase orographic clouds**
**during 2015 ACAPEX field campaign**
Yun Lin[1,2], Jiwen Fan[1,*], Pengfei Li[3,4], L. Ruby Leung[1], Paul J. DeMott[5], Lexie Goldberger[1],
Jennifer Comstock[1], Ying Liu[1], Jong-Hoon Jeong[1,2], Jason Tomlinson[1]
[1] Atmospheric Sciences and Global Change Division, Pacific Northwest National Laboratory,
Richland, WA 99352, USA
[2] Joint Institute for Regional Earth System Science and Engineering (JIFRESSE), University of
California, Los Angeles (UCLA), Los Angeles, CA 90064, USA
[3] College of Science and Technology, Hebei Agricultural University, Baoding, Hebei
14 071000, P.R. China
[4] Research Center for Air Pollution and Health; Key Laboratory of Environmental
Remediation and Ecological Health, Ministry of Education, College of Environment and
Resource Sciences, Zhejiang University, Hangzhou, Zhejiang 310058, P.R. China
[5] Department of Atmospheric Science, Colorado State University, Fort Collins, CO 80523, USA
Corresponding author: Jiwen Fan (jiwen.fan@pnnl.gov)



**Abstract**
A large fraction of annual precipitation over the western United States comes from wintertime
orographic clouds associated with atmospheric rivers (ARs). The transported African and Asian
dust and marine aerosols from the Pacific Ocean may act as ice-nucleating particles (INPs) to
affect cloud and precipitation properties over the region. Here we explored the effects of INPs
from marine aerosols on orographic mixed-phase clouds and precipitation at different AR stages
for an AR event observed during the 2015 ACAPEX field campaign under low dust ($< 0.02$ cm$^-$
$^3$) conditions. Simulations were conducted using the chemistry version of the Weather Research
and Forecasting model coupled with the spectral-bin microphysics at 1-km grid spacing, with ice
nucleation connected with dust and marine aerosols. By comparing against airborne and ground-
based observations, accounting for marine INP effects improves the simulation of cloud phase
state and precipitation. The marine INPs enhance the formation of ice and snow, leading to less
shallow warm clouds but more mixed-phase and deep clouds, and increased ice water path (over
5 times) and snow precipitation (over 40 times). The responses of cloud and precipitation to
marine INPs vary with the AR stages with more significant effects before AR landfall and post-
AR than after AR landfall, mainly because the moisture and temperature conditions change with
the AR evolution. This work suggests weather and climate models need to consider the impacts
of marine INPs since their contribution is notable under low dust conditions despite the much
lower relative ice nucleation efficiency of marine INPs.



**1 Introduction**

Atmospheric river (AR) events have great impacts on atmospheric and hydrological

processes in the western United States during winter. They account for 30–50% of the total
winter precipitation through their impacts on orographic clouds and associated heavy
precipitation (Dettinger et al., 2011). Understanding the factors influencing different types of
precipitation (rain vs. snow) associated with ARs is crucial for planning and managing regional
water resources and hydrologic hazards and improving atmospheric and hydrologic forecasting
in the western United States. Rain and snow precipitation produced by orographic clouds over
the Sierra Nevada Mountains is closely related to the partitioning between cloud liquid and ice
phases, which can be largely modified by aerosol particles (Rosenfeld et al., 2013; Fan et al.,
2014, 2017b). However, aerosol-orography-precipitation relationships are complicated,
depending on aerosol properties, mountain geometry, cloud phase, temperature, humidity, and
flow patterns as reviewed in Chouldhury et al. (2019).

Over the western United States, understanding the roles of aerosols, particularly those

capable of initiating ice crystal formation, in altering clouds and precipitation is still limited,
which has motivated recent observational and modeling studies (Ault et al., 2011; Creamean et
al., 2013, 2015; Rosenfeld et al., 2013; Fan et al., 2014, 2017b; Martin et al., 2019; Levin et al.,
2019). While it has been found that long-range transported aerosols particularly dust particles as
ice nucleating particles (INPs) influence clouds and precipitation in the mountainous western
United States (Uno et al., 2009; Ault et al., 2011; Creamean et al., 2013), it is also clear from
measurements that clouds occurring in and around ARs can also be influenced by INPs with
apparent sources from the ocean (Levin et al., 2019).



Where sufficient INPs are present, heterogeneous ice nucleation can occur at a higher
temperature where liquid and ice can co-exist, in contrast to homogeneous freezing which
normally requires -38 °C or colder environment (DeMott et al., 2010; Vali et al., 2015).
Increasing INPs can lead to stronger ice formation and growth at the expense of supercooled
liquid and increase precipitation. This concept is the basis of using cloud seeding to increase
orographic precipitation (Reynolds, 1988; Geerts et al., 2010; French et al., 2018). For
orographic clouds in the western United States, previous studies showed that INPs can increase
total precipitation through the "seeder feeder" mechanism (Choularton and Perry, 1986;
Creamean et al., 2013), in which ice crystals that form in the upper portions of orographic clouds
can collect droplets and grow to a larger size as they fall through a supercooled liquid layer
before reaching the ground. Fan et al. (2014, 2017b) found that INPs like dust particles can
increase precipitation by enhancing riming and deposition processes in mixed-phase orographic
clouds, consistent with other studies (e.g., Muhlbauer and Lohmann, 2009; Xiao et al., 2015;
Hazra et al., 2016; Yang et al., 2020). Fan et al. (2017a) also noted that the relative importance of
riming to deposition depends on the mixed-phase cloud temperatures. Despite the importance of
INPs in cloud formation and precipitation, they typically have a low abundance and large
variations in their nucleating characteristics, especially in terms of the temperatures over which
they initiate ice crystal formation (Kanji et al., 2017; Levin et al., 2019). Hence, there is large
uncertainty in evaluating INPs impacts on mixed-phase and ice clouds as well as precipitation.
It is known that dust particles are important INP sources, which can initiate freezing over
a range of temperatures but most efficiently below −20 °C (Murray et al., 2012; Kanji et al.,
2017). Another important type of INPs is terrestrially sourced biological particles, which cause
freezing at temperatures as warm as −5 °C (Murray et al., 2012). During ARs, the long-range



transport of dust or biological particles is highly episodic. Sea spray or marine aerosols
consisting of sea salt and marine organic carbon resulting from wave breaking and bubble
bursting at the ocean surface may also be a source of INPs (Burrows et al., 2013; Vergara-
Temprado et al., 2017; McCluskey et al., 2018b; Levin et al., 2019). Recently, McCluskey et al.
(2018a) derived an ice nucleation parameterization for INPs from sea spray aerosols based on
observations collected at a North Atlantic coastal site and its relation to the marine aerosol
surface area. Given the distinct physio-chemical characteristics and the different ice-nucleating
efficiency, the impact of marine INPs on cloud and precipitation could be very different from
dust or biological particles (DeMott et al., 2016; Kanji et al., 2017). However, studies of marine-
sourced INP effects on clouds and associated precipitation are limited (Kanji et al., 2017; Levin
et al., 2019). A few previous studies investigated the impacts of marine INPs on precipitation and
radiation with global climate models (Hoose et al., 2010; Burrows et al., 2013; Yun and Penner,
2013), albeit without the advantage of direct data on their ice nucleation efficiencies. Further, a
detailed, process-level understanding of how marine INPs affect mixed-phase cloud processes
and precipitation is lacking.

Following the CalWater campaigns in 2009, 2011, 2014, an interagency sponsored study,

CalWater 2015, utilized a larger suite of instruments and measurement platforms to study ARs
and aerosol-cloud interactions in AR environments (Ralph et al., 2016). As part of CalWater
2015, the U.S. Department of Energy sponsored Atmospheric Radiation Measurement (ARM)
Cloud Aerosol Precipitation Experiment (ACAPEX) field campaign aimed specifically at
improving understanding and modeling of aerosol impacts on winter storms associated with
landfalling ARs (Leung et al., 2016). The ACAPEX campaign conducted intensive sampling of
clouds and aerosols using instruments on board the ARM Aerial Facility Gulfstream (G-1)





aircraft and ARM Mobile Facility on board the research vessel Ron Brown. These measurements
were made in conjunction with clouds and aerosols, meteorological, hydrological, and oceanic
measurements collected by instruments on three other aircraft and Ron Brown and at a coastal
surface station. Collectively, these data provide a unique opportunity to examine the complex
interactions among aerosols, orographic clouds, and ARs.

A major AR event spanning over 5 - 9 February 2015 occurred during the ACAPEX

campaign and made landfall on the coast of Northern California, producing heavy rainfall with
some regions receiving up to 400 mm of total precipitation during the event (Ralph et al., 2016;
Cordeira et al., 2017). This AR event was extensively sampled by the (G-1 aircraft (Schmid et
al., 2014) for characterizing aerosol and cloud properties. During this event, marine aerosols
were the main aerosol type and marine INPs were dominant at cloud activation temperatures.
Aerosol sampled by G-1 indicated that dust and biological particles were rather scarce in and
around ARs, which is in stark contrast to the dominance of dust INPs during the AR events in the
CalWater 2011 campaign (Levin et al., 2019). Therefore, the AR event during the ACAPEX
campaign provides a rather unique opportunity to explore the role of marine aerosols in the
orographic clouds and precipitation associated with landfalling ARs in the western United States.

In our previous modeling studies (Fan et al., 2014, 2017b), we implemented an

immersion freezing parameterization for dust particles (DeMott et al. 2015) in a spectral-bin
microphysics (SBM) scheme to examine the long-range dust effects on AR-associated
orographic mixed-phase clouds and precipitation during CalWater 2011. With marine INPs
dominating in CALWATER 2015/ACAPEX, in this study we implemented the recently
developed ice immersion nucleation parameterization for sea spray aerosols by McCluskey et al.
(2018b) in the SBM scheme. To explicitly simulate various aerosol types, different from Fan et



al. (2014, 2017a) who prescribed aerosols based on observations, a chemistry version of the
Weather Research and Forecasting model (WRF-Chem) coupled with the SBM (Gao et al.,
2016) was employed to predict aerosol properties and their interactions with clouds and radiation
for the AR event on 6 - 9 February 2015. We focused on exploring the effects of INPs from sea
spray aerosols, in competition with mineral dust INPs, on the orographic mixed-phase clouds and
precipitation at different stages of the AR event as thermodynamic conditions evolved with the
different AR stages.
**2 Model configuration and experiment design**

The WRF-Chem version 3.6 coupled with SBM as described in Gao et al. (2016) is

employed for model simulations of this study, in which SBM is coupled with the Model for
Simulating Aerosol Interactions and Chemistry (MOSAIC; Fast et al., 2006; Zaveri et al., 2008).
The SBM scheme is a fast version in which ice crystal and snow (aggregates) in the full version
are represented with a single size distribution (low-density ice) with a separation at 150 μm in
radius, with graupel or hail populating larger sizes (Khain et al., 2009, 2010; Fan et al., 2012,
2017a). Here we choose the graupel version since hail is not one of the major cloud
hydrometeors in the case we simulate. The WRF-Chem-SBM model is particularly designed to
improve simulations of aerosol effects on clouds for complicated aerosol compositions and
heterogeneous spatial distribution of aerosols. It has been applied in several studies including
warm stratocumulus clouds (Gao et al., 2016), thunderstorms (Fan et al. 2020; Zhang et al.,
2020), and supercell storms (Lin et al., 2020). Here WRF-Chem-SBM is employed, different
from our previous studies in Fan et al. (2014, 2017a) which used WRF-SBM with prescribed
aerosols, in order to explicitly simulate various aerosol types including marine aerosols and dust
particles.



The four-sector MOSAIC aerosol module is chosen for the simulations of aerosols and
the CBMZ (Carbon Bond Mechanism version Z) is used for gas-phase chemistry. The MOSAIC
module treats nine major aerosol species (sulfate, nitrate, chloride, ammonium, sodium, black
carbon, primary organics, other inorganics (OIN), and water). OIN is used as a surrogate of dust
and the production of dust is parameterized with the dust transport model DUSTRAN (Shaw et
al., 2008). Sea salt aerosol (the combination of sodium and chloride), as a surrogate for all SSA,
is parameterized as a function of sea-surface wind speed  (Gong et al., 1997b, a). The dry
diameters of the particles over the four bins have a range of 0.039–0.156, 0.156–0.624, 0.624–
2.5, and 2.5–10.0 μm, respectively. For the total aerosol, aerosol size distribution over each
section is represented with a 2-moment approach that predicts aerosol mass and number
following a log-normal distribution [Simmel and Wurzler, 2006]. For each composition such as
dust and sea salt, only the mass mixing ratio in each section is predicted and outputted. The
aerosol number mixing ratio in each bin is only predicted for the total aerosol. Therefore, in this
study, the dust and sea salt number mixing ratios used for ice nucleation parameterizations are
derived based on their respective mass mixing ratio by assuming the same size and density of all
particles over each bin, that is,
$$N_{i,j} = \frac{m_j}{6\pi (D_j)^3 \rho_i}$$

where $i$ denotes the aerosol composition (sea salt or dust here), $j$ denotes the $j^{th}$ aerosol bin, $m_j$ is
the total mass mixing ratio of the $j^{th}$ bin, $\rho_i$ is the assumed density (i.e., 2.6 g cm$^{-3}$ for dust and
2.2 for sea salt), and $D_j$ is the geometric mean diameter of the $j^{th}$ bin. The approach for deriving
the number mixing ratio for each aerosol component has been used in the literature (i.e., Zhao et
al., 2013). We understand that the assumption that all particles have the same size over each bin



may introduce some uncertainty. However, the size distribution of each aerosol component is
unknown in the model and any assumption on the size distribution might introduce uncertainty.

**2.1 Implementing an immersion freezing parameterization for marine INPs**

In the original SBM model, the ice nucleation accounting for both deposition ice

nucleation and condensation-freezing is parameterized based on Meyers et al. (1992) and Bigg
(1953) is employed for the immersion and homogeneous drop freezing. Neither of the ice
nucleation parameterizations is connected with aerosols. Bigg (1953) was formulated based on
the stochastic hypothesis where the freezing probability is assumed proportional to drop mass
and the freezing rate is as a function of temperature without involving INPs. Fan et al. (2014,
2017a) implemented DeMott et al. (2015) as an immersion freezing parameterization to
investigate the effects of dust INPs on orographic mixed-phase clouds and precipitation during
CalWater 2011. We adapted this implementation to WRF-Chem-SBM for this study to connect
ice nucleation with dust particles. Developed based on both laboratory data and field
measurements, DeMott et al. (2015) is an empirical parameterization for immersion freezing of
natural mineral dust particles. INP concentrations are quantified as functions of temperature and
the total number concentration of particles larger than 0.5 μm diameter. In our implementation,
the dust number mixing ratio for each aerosol bin is derived from its mass as detailed in the
section above. The total dust number mixing ratio inputted to DeMott et al. (2015) is the
integration over 0.5 -10 μm.

To connect ice nucleation with sea spray aerosols, we implemented McCluskey et al.

(2018a, thereafter MC2018), which was developed for quantifying ice nucleating activity by
marine organics over the North Atlantic Ocean, in SBM following a similar approach as the
implementation of DeMott et al. (2015).  The nucleation site density in MC2018 is described as





$$n_s = exp(-0.545(T - 273.15) + 1.012)$$

where $n_s$ is the nucleation site density (m$^{-2}$) and $T$ is the temperature (K). With $n_s$ determined by
MC2018, the nucleated ice particle concentration is obtained following Niemand et al. (2012) as
$$\sum_{j=1}^{n} N_j = \sum_{j=1}^{n} N_{\text{tot},j}\{1 - \exp[-S_{\text{ae},j}n_s(T)]\}$$


where $S_{ae,j}$ is the surface area of individual sea spray aerosol particles in the $j^{th}$ bin which is
calculated from $\pi D_j^2/4$ ($D_j$ is the geometric-mean diameter), $N_{tot,j}$ is the total sea spray aerosol
number in each bin which is derived from its mass as detailed in the section above, and $N_j$ is the
ice particle number in each bin. Sea salt particles are used as the surrogate of sea spray aerosols
given that most marine organic aerosols exist with coating on the surface of sea salt particles in
the size range that dominates surface area (e.g., Prather et al., 2013). MC2018 can have efficient
ice nucleation at warm temperatures like -15 °C or warmer.

Bigg et al. (1953) is employed only for homogeneous drop freezing when the temperature

is colder than -37 °C. As discussed in Fan et al. (2014), the deposition-condensation freezing is
turned off because the simulation with deposition-condensation freezing produces a large
number of small ice particles, which is not consistent with the observed mixed-phase cloud
properties in the study region. Contact freezing is also turned off due to the negligible
contribution (Fan et al., 2014).

**2.2 Experiment design**

Simulations are configured with two nested domains using the nesting down approach

(i.e., the inner domain is run separately driven by the outer domain), covering most of the





western US (Fig. 1). The outer domain consists of 399 × 399 grid points with a horizontal grid
spacing of 3 km and the inner domain consists of 498 × 390 grid points with a horizontal grid
spacing of 1 km. 50 vertical levels with stretched intervals are configured, with a grid spacing of
70 m at the lowest levels and ~400 m at the model top. The dynamics time step is 15 seconds for
the outer domain and 5 seconds for the inner domain.

The simulation for the outer domain starts at 00:00 UTC on February 3 and runs for 48

hours for chemistry spin-up using the WRF-Chem-SBM model, driven by global WRF-Chem
simulation as the initial and boundary conditions of gas-phase species and aerosols and the
Modern-Era Retrospective analysis for Research and Applications, Version 2 (MERRA2; spatial
resolution of 0.5 by 0.5 degree and temporal resolution of 6-hourly) as the initial and boundary
conditions of meteorological fields. Then the outer domain simulation is reinitialized at 00:00 UTC
on February 5 using the meteorological data from MERRA2 to avoid the large error growth in
meteorology associated with long-time model integration, although the chemistry simulations is a
continuation from the spin-up run, and runs until 23:00 UTC on February 8. Given that running
the WRF-Chem-SBM fully-coupled model is extremely computationally expensive for 1-km grid
spacing in the inner domain, we interpolate aerosol-related quantities such as aerosol composition,
hygroscopicity, and mass and number concentrations from the outer domain simulations using
bilinear interpolation for the inner-domain simulation to reduce computational cost. This means
we conduct the inner-domain simulation separately with chemistry turned off, and aerosol
information is updated hourly using data from the outer domain simulations. The inner-domain
simulation is run from 00:00 UTC on February 5 to 23:00 UTC on February 8, and the initial and
boundary meteorological conditions are from MERRA2. To validate this approach, we compare
the simulation with fully coupled WRF-Chem-SBM for the inner domain simulation and found

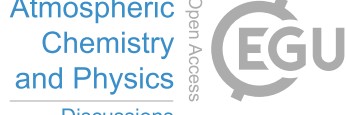

that the two simulations resemble each other in terms of precipitation (Fig. S1). Therefore, it is a
valid approach that saves computation time by about 40%.

For emissions data, the U. S. Environment Protection Agency (EPA) National Emission

Inventory (NEI) with a 4 km by 4 km horizontal resolution based on the year 2011 rates
(NEI2011) is commonly used for anthropogenic emissions in the United States. However, using
NEI2011 predicts too large anthropogenic aerosol mass compared with observations. Since the
emissions of gaseous species and particulate matter decreased significantly from 2011 to 2015 in
California (Table S1), the California Air Resources Board emission inventory in 2015
(CARB2015) is used for anthropogenic emissions input for California, while NEI2011 is used
for other states in the simulation domain. The use of NEI2011 for other states is acceptable since
the lower and middle atmosphere in the simulation domain is dominated by southwesterly winds
during the simulation period that transport air pollutants from coastal to inland regions. The use
of CARB2015 reduces the simulation of aerosol number concentrations mainly below 2 km by
40% relative to the use of NEI2011, in better agreement with observations.

The Model of Emissions of Gases and Aerosols from Nature (MEGAN) with a monthly

temporal and 1 km horizontal resolution (Guenther et al., 2012) is used for biogenic emissions.
The Rapid Radiative Transfer Model for application to GCMs (RRTMG) is used for shortwave
and longwave radiation schemes (Iacono et al., 2008), the Noah Land Surface Model for land
surface physics (Chen and Dudhia, 2001), and the Mellor-Yamada-Janjic (MYJ) scheme for
planetary boundary layer parameterization (Mellor and Yamada, 1982; Janjić and Prediction,

2001).

Three simulations were carried out over the inner domain for this study to investigate the

impacts of marine INPs: (1) The reference case is Bigg, using the default immersion freezing



parameterization of Bigg et al. (1953) in SBM which is temperature-dependent only; (2)
DM15+MC18, in which both DeMott et al. (2015) and MC2018 parameterizations are used for
ice nucleation from dust and marine aerosols, respectively; (3) DM15, using the parameterization
of DeMott et al. (2015) for dust aerosols (diameter > 0.5 µm) with MC2018 turned off. The
impacts of marine INPs are derived by comparing the DM15+MC18 and DM15 simulations.
**3 Results**
**3.1 Model evaluation with observations**

We evaluate the model simulations of aerosol and cloud properties and surface

precipitation. Figure 2a shows a comparison of modeled aerosol properties including aerosol
number concentration and chemical composition from the simulation of DM15+MC18 intended
to represent the observed case, with the G-1 aircraft measurements on 7 February. Aerosol
properties in all three simulations are similar, and thus only DM15+MC18 is shown. Overall, the
simulated aerosol number concentration over the size range of 0.067 - 3 µm is comparable to the
observations over the same size range estimated by combining data from the Ultra-High-
Sensitivity Aerosol Spectrometer (UHSAS) and the Passive Cavity Aerosol Spectrometer Probe
(PCASP) at below 2-km altitude. The simulation overestimates the total aerosol number
concentrations up to ~ 2-times at altitudes of 2.2-3.2 km. For aerosol composition, the airborne
Aerosol Time of Flight Mass Spectrometry (ATOFMS) measurements provided the mean
fractional number contributions of aerosol source classifications (Levin et al., 2019), which is
shown in Fig. 2b. For comparison with the model, the mean mass contributions of the
corresponding aerosol source classifications are computed since the number concentrations of
individual aerosol components are not predicted by WRF-Chem (Fig. 2c). Both the observed
fractional number contributions and the simulated mass contributions show that marine aerosols


are dominant during the AR event, accounting for more than 60% of the total aerosol number
based on ATOFMS measurements and total aerosol mass based on the simulation. Although the
simulated dust mass fraction is ~14%, the derived number concentration is very low (less than
$0.02$ cm$^{-3}$ for sizes larger than 0.5 μm, shown in a later figure). This is because dust is mainly
from aerosol bins at larger sizes. The number concentrations of the sea salt aerosols are generally
three orders of magnitude higher than those of dust, and these numbers populate smaller bins of
the aerosol distribution (97% from the first two aerosol size bins) even though the sea salt mass
is predominately at larger sizes (96% from the last two size bins).

Figure 3 presents an evaluation of precipitation, showing the accumulated precipitation

during the AR event from 06:00 UTC 5 February to 09:00 UTC 8 February 2015 (Fig. 3a) and
the time-series of mean precipitation rates averaged over the observation stations (Fig. 3b). The
observed precipitation rates are from the rain gauge measurements, provided by the NOAA Earth
System Research Laboratory's Physical Sciences Division
(https://psl.noaa.gov/data/obs/datadisplay). The model generally captures the spatial pattern of
the observed accumulated precipitation (Fig. 3a) and reproduces the temporal evolution of the
precipitation rate (Fig. 3b). Two major precipitation periods in the observations including the
AR-induced orographic precipitation and the postfrontal precipitation are generally captured by
the simulations, although the simulated postfrontal precipitation occurs several hours later in the
simulations compared to the observations. The peak values of precipitation are overestimated by
all the simulations (25-50%) relative to the observations (Fig. 3b), but the values from
DM15+MC18 are lower than the other two, closer to the observations. From the spatial
distributions (Fig. 3a), the observed accumulated precipitation in the southern part of the
mountain range is generally less than 100 mm. DM15+MC18 simulates it well, whereas the


other two simulations overestimate precipitation in that area. The lower precipitation by
DM15+MC18 is mainly because of the spillover effect caused by marine INPs (Fig. 3a, right).
That is, with marine INPs, more ice/snow formed over the windward side falls slower than rain
and more of them are transported to the lee side. In the northern part of the domain (> 40º N),
DM15+MC18 predicts more precipitation than the other two simulations. The simulated
precipitation between Bigg and DM15 are very similar, suggesting that in a low dust
environment, the temperature-dependent Bigg (1953) parameterization simulates similar ice
formation as DeMott et al. (2015).

Cloud phase is crucial to radiation and precipitation for mixed-phase clouds, and the

glaciation ratio is usually used to represent the cloud phase states. The glaciation ratio is defined
as (IWC/(IWC+LWC)), where LWC and IWC denote liquid and ice water content, respectively.
Values less than 0.1 and larger than 0.9 denote the liquid phase and ice phase, respectively, with
values between 0.1 and 0.9 for the mixed-phase (Korolev et al., 2003). The G-1 aircraft sampled
the postfrontal clouds on February 7 as shown in Fig. 4a. All three simulations cannot capture the
observed size of the precipitation cell (Figs. 4b and S2).  In the simulations, precipitation is
dominated by a few heavy precipitation clusters instead of the observed wide precipitation area.
The simulated cells also do not reach the high altitudes found in the observations. The deviations
of the simulation from observations for the postfrontal clouds could be mainly because of the
long-time model integration time (the 4[th] day after model initiation). Anyhow, DM15+MC18
simulates the largest size of the precipitation cell, with the highest vertical extent among the
three simulations.

The LWC and IWC are derived from observations with the Water Content Monitor

(WCM), an instrument that uses the impact of water on several heated wires as the basis for





measuring cloud total water content (TWC) and liquid water content (LWC) from which the ice
water content (IWC) can be derived [Baumgardner et al., 2011; Matthews et al., 2015]. LWC and
IWC along both horizontal and vertical flight segments are displayed in Fig. 5a-b. IWC is
generally 2-4 times larger than LWC in the postfrontal clouds. To compare with observations,
the model data are processed by: (a) selecting the grids at a distance from the simulated cell
center similar to the distance of the airplane position from the observed postfrontal cell center,
and sampling the data at a similar ambient temperature as observed by the aircraft (around -10 ºC
along the horizontal segment shown in Fig. 5a); (b) accounting for the location mismatch and
increasing the sample size in the simulation to be more representative by extending the sampling
area to include 20 grids at the front and back of a selected grid along the flight track, mimicking
approximately the distance traveled by the G-1 airplane in five minutes; (c) filtering out the
sampled grids with values of (LWC + IWC) below the detection limit of WCM (i.e., 0.02 g m$^{-3}$,
Thompson et al., 2016). Both horizontal and vertical flight segments are incorporated for
comparison.
Figure 5 shows comparisons of LWC and IWC and the glaciation ratio of
IWC/(IWC+LWC) between the simulations and aircraft measurements. LWC is overestimated in
all three simulations, while IWC is underestimated in Bigg and DM15 but overestimated in
DM15+MC18 (Fig. 5c). The mean glaciation ratios fall in the range of 0.1- 0.9 among the
simulations (Fig. 5d), indicating that the observed mixed-phase cloud feature is simulated by the
model. DM15+MC18 shows a mean ratio of ~ 0.70, similar to the observed value of 0.74. This
shows that the mixed-phase state is well captured when the marine INP effect is considered. In
contrast, in Bigg and DM15 with a glaciation ratio is 0.41 or less, the mixed-phase state is liquid-
dominated. The inclusion of the marine INP effect improves the simulation of cloud phase states



via enhancing heterogenous ice formation through immersion freezing. A detailed examination
of how the marine INPs impacts ice nucleation and cloud properties will be discussed in the
following section.

**3.2 Marine INP effects under different AR stages**

Impacts of the marine INPs transported from the Pacific Ocean on orographic clouds and

precipitation are revealed by comparing the simulation of DM15+MC18 with the simulation of
DM15.

The AR evolution has three distinct stages: before AR landfall (from 06:00 UTC 5 to

18:00 UTC 6 February), after AR landfall (from 18:00 UTC 6 to 12:00 UTC 7 February), and
post-AR (from 12:00 UTC 7 to 09:00 UTC 8 February). The three stages can be identified from
the change of the integrated water vapor (IWV) with time during the event (Fig. 6a). Before AR
landfall, IWV in most of California was relatively low (Fig. 6a, left). IWV in northern California
increased as the AR made landfall at about 18:00 UTC on 6 February and brought ample water
vapor to California (Fig. 6a, middle). Heavy orographic precipitation along the Sierra Nevada
Mountains occurred during this period (Fig. 7a). At 12:00 UTC 7 February, the AR started to
retreat (Fig. 6a, right), and postfrontal cloud cells formed, with relatively small cloud fraction
and precipitation (Fig. 7a).

Vertical profiles of the thermodynamic and kinematic environments at the three stages

are shown in Fig. 6b. The thermodynamic and kinematic environments significantly varied with
the AR stages. After AR landfall, water vapor increased significantly in the lower atmosphere
(below 5 km), but the middle and upper levels became drier (dashed, Fig. 6b) compared with the
stage before AR landfall (solid). The vertical motion also weakened after AR landfall (Fig. 6d),





suggesting that the atmosphere became more stable. At the post-AR stage, moisture above 2-km
altitude was reduced compared to after AR landfall. Note that the temperature below 8 km was
colder by up to 6 °C at the post-AR stage compared to the previous two stages (Fig. 6c). These
differences in the meteorological conditions among the different stages are very important to
understand the cloud and precipitation properties and their responses to marine INPs. The mean
water vapor and temperature profiles are similar between DM15 (blue) and DM15+MC18 (red),
as seen from the overlapping blue and red lines.

From the time series of average precipitation rates (Fig. 7a), the effect of marine INPs

varies with the different AR stages, from substantial increases of precipitation (up to 330%)
before AR landfall (the red dotted line, second y-axis) to no significant effects (a very small
increase) after AR landfall. In the first stage (before AR landfall), the total precipitation increases
by 36% on average due to the marine INP effect (Fig. 7a and Table 1). There is only a 4%
increase in the total precipitation after AR landfall. The total precipitation at the post-AR stage is
small and the change from DM15 and DM15+MC18 is negligible. Thus, the marine INP effect
only significantly increases the total precipitation over the domain at the stage before AR landfall
when a moderate amount of precipitation occurs in northern California (Fig. 8a). After AR
landfall, precipitation increases significantly. Although the total precipitation is not changed
much by the marine INPs, the marine INPs produce a spillover effect featuring reduced
precipitation on the windward slope of the mountains but increased precipitation over the lee side
(Fig. 8b and Fig. 9e). This is because with the marine INPs, the larger amount of ice/snow that
forms on the windward slope is transported to the lee side (Fig. 9d) and grows to a larger size
and precipitates as snow. This spillover effect is accompanied by a large reduction of cloud water
and rain over the windward side because of conversion of liquid to ice (Fig. 9b-c). Since the



water vapor transport along the cross-section is very similar between DM15 and DM15+MC18
(Fig. 9a), the spillover effect by marine INPs is mainly the result of different cloud
microphysical properties instead of meteorological conditions.

Even though the total domain precipitation is not changed much by the marine INPs at

the later two stages, the cloud phase and the near-surface precipitation type (i.e., rain or snow)
are notably changed (Table 1).  The mean glaciation ratio in the mixed-phase is very low in
DM15 (0.14, 0.16, and 0.001 for the 1st, 2nd, and 3rd stages, respectively) and is increased to 0.74,
0.59, and 0.36, respectively. We examine the ratio of snow/(rain+snow) in mass mixing ratio at
the lowest model level for the changes of the near-surface precipitation type (Fig. 7b).  There is
negligible snow precipitation near the surface in DM15 and the ratios of snow precipitation are
very small during the entire AR event. The snow precipitation ratios increase in DM15+MC18
and the magnitudes vary significantly with different AR stages. On average, the ratio of snow
precipitation increases from 0.002, 0.001, <0.001 in DM15 to 0.08, 0.04, and 0.13 in
DM15+MC18 before AR landfall, after AR landfall, and post-AR, respectively (Table 1). This
shows that the marine INPs increase snow precipitation and the effect is particularly significant
before AR landfall and post-AR. Correspondingly, rain precipitation is reduced (Table 1). This
has an important implication for the regional hydrological resource since more snow
accumulation in winter increases freshwater resources in the summer while less rain reduces
flood risks.

The increased snow and reduced rain at the surface correspond to the increased ice water

path (IWP) and decreased liquid water path (LWP; Fig. 7c). The mean LWP in DM15+MC18 is
reduced by 66%, 46%, and 26% for the three stages relative to DM15, respectively (Table 1). We
showed an increased LWC from DM15 to DM15+MC18 in Figure 5c in the postfrontal cells.



Here the decrease in LWC/LWP averaged over the entire post-AR stage is dominated by the
strong decrease over the time before the postfrontal cloud formed. The postfrontal cells are
invigorated (see section 3.3) by marine INPs so both LWC and IWC are increased as shown in
Figure 5. IWP is greatly enhanced by about 8, 5, and 440 times at the three stages, respectively.
Interestingly, the total condensate water path (TWP) is increased by the marine INPs (Fig. 7d).
On average there are 45%, 29%, and 35% increases in TWP in DM15+MC18 at the three AR
stages relative to DM15, respectively (Table 1). The increases in the total condensate water path
and the increased surface precipitation (or no change) suggest that marine INPs enhance the
conversion of water from the vapor phase to the condensate phase, which will be further
discussed later. This is particularly the case before AR landfall with water vapor content notably
reduced in DM15+MC18 compared with DM15 (Fig. S3a).

Cloud cover is slightly increased during the first two stages (4-5%) in the simulations

including marine INPs, but the change at the post-AR stage is ~ 20% on average, which is very
significant. Because both TWP and cloud cover are increased due to the marine INP effect, the
cloud radiative forcing (CRF) at TOA gets stronger by 15%, 13%, and 10% for the three AR
stages, respectively. Although the cloud phase, precipitation type, and cloud fraction at the post-
AR stage have the largest changes among the three stages by the marine INP effect (Table 1), the
CRF does not change drastically probably because of the offset between the increase resulting
from the increased cloud fraction and TWP and the decrease from the reduced cloud liquid is the
largest.

Overall, the marine INP effects on TWP, IWP, and snow precipitation are more

significant at the first and third stages (i.e., before AR landfall and post-AR) than the stage after
AR landfall. But a notable spillover effect is seen after AR landfall.  The cloud and precipitation





quantities are more sensitive to marine INPs before AR landfall than after AR landfall, and the
responses of TWP/IWP and snow precipitation are particularly drastic at the post-AR stage
(Table 1). The reasons leading to the different responses at different AR stages are now
examined.

**3.3 Explaining different marine INP effects at different AR stages**

We first examine the temporal evolution of dust and marine aerosol number

concentrations, which are derived based on the predicted mass mixing ratios as described in
Section 2 and used as input to the DeMott et al. (2015) and MC2018 parameterizations (Fig. 10a,
b), as well as their corresponding immersion freezing (i.e., ice nucleation) rates (Fig. 10c, d). The
dust concentrations and the corresponding ice nucleation rates (Fig. 10a, c) are about three orders
of magnitude lower than those of the marine aerosols (Fig. 10b, d) during the AR events. This is
driven both by the activation temperature spectrum of dust and its very low mass/number
concentrations in this case. Ice nucleation from dust is negligible at temperatures warmer than -
15 ºC but the ice nucleation from marine aerosols is notable. This is mainly because of three
orders of magnitude higher marine aerosol number concentrations from the surface up to 8 km.
The deep marine aerosol layer during the AR allows notable ice nucleation at temperatures
higher than -15 ºC. The clear-sky marine aerosol number concentrations increase from before
AR landfall to post-AR as the AR evolved (Fig. 10b). After the AR makes landfall, marine
aerosols increase significantly as AR strong winds near the ocean surface produce them more
and also transport more to the Sierra Nevada Mountains (Fig. 10b). Despite the significant
increase in marine aerosols after AR landfall, the marine INP effects on clouds and precipitation
are small at this stage, because the increase of marine aerosols does not increase ice nucleation



rates (Fig. 10d). However, at the post-AR stage, the ice nucleation rates from the marine INPs
are up to a few times larger than the earlier two stages (Fig. 10d), explaining why the effects on
IWP and snow precipitation at the post-AR stage are largest among the three stages.

To further understand how and why the cloud and precipitation responses to marine INPs

are different at different AR stages, we separate clouds into three cloud regimes: a shallow warm
cloud regime with cloud top temperature (CTT) warmer than 0 °C, a mixed-phase cloud regime
with CTT between -30 and 0 °C, and a deep cloud regime having CTT colder than -30 °C and
cloud base temperatures above 0 °C.  Figure 11 shows that the marine INP effect consistently
shifts the cloud occurrences from the shallow warm cloud regime to mixed-phase and/or deep
cloud regimes among the three AR stages. It is noted that the deep cloud regime is enhanced
much more at the first and third stages than the second stage, i.e., 22% before AR landfall and
235% at the post-AR stage but only 8% after AR landfall. The post-AR stage also has the largest
increase in mixed-phase cloud occurrences.

Accordingly, the mean cloud depth for each cloud regime is changed by marine INPs,

with a decrease for the shallow warm clouds and an increase for the mixed-phase and deep
clouds (Fig. 11b). Before AR landfall, the increase in the deep cloud depth is largest while at the
post-AR stage, the increase in the mixed-phase cloud depth is the largest.  Consistent with a shift
in cloud regimes, the total precipitation produced by shallow warm clouds is reduced by 9%,
22%, and 16% while the total precipitation produced by deep clouds is increased by 66%, 4%,
and 350%, respectively, at the three AR stages (Fig. 11c).  Therefore, the large increase in the
surface accumulated precipitation by marine INPs before AR landfall (36%) is mainly because of
the increase in deep cloud precipitation. The larger occurrence of deep clouds at this stage is
consistent with a larger increase in TWP and reduction in moisture. Although the relative



increases in deep cloud occurrences and precipitation by marine INPs are very large at the post-
AR stage, their occurrences are so small that their contribution to the total precipitation is
negligible.

How do marine INPs reduce shallow warm clouds but invigorate mixed-phase and deep

clouds and why is this effect larger at the first and third stages? Marine INPs greatly enhance ice
and snow number concentrations and mass mixing ratios through immersion freezing, which
converts drops to ice or snow particles (Figs. 12a and 13a).  The mean number concentrations
and mass mixing ratios of ice particles (ice +snow) in mixed-phase and deep cloud regimes are
several orders of magnitude higher in DM15+MC18 than in DM15. As detailed in Fan et al.
(2017a) which studied the same type of mixed-phase clouds in the same region, more ice/snow
particles forming from the immersion freezing enhance the Wegener–Bergeron–Findeisen
(WBF) and riming processes (Table 2), converting supercooled drops to ice or snow and leading
to more ice/snow but fewer cloud droplets and raindrops (Figs. 12b, c and 13b, c). The
reductions of cloud droplet and raindrop number concentrations and mass mixing ratios from
DM15 to DM15+MC18 are larger before AR landfall and during post-AR relative to the stage
after AR landfall, corresponding to a larger shift to the mixed-phase and deep clouds. Thus, the
larger increases in deposition/WBF and riming rates are seen (Table 2).

As discussed earlier, the largest ice nucleation rates at the post-AR stage explain the

largest marine INP effects among the three stages. The postfrontal clouds have the lower cloud
top heights (warmer than -25 ºC, i.e., shallower clouds) compared with the clouds at the first two
stages, thus the dust INP nucleation rates are smaller (negligible) as shown in Fig. 10c but the
deep marine aerosol layer and its action as INPs adds significantly to ice nucleation. In addition,
with the ~ 6 ºC colder temperatures below 8-km altitudes during the post-AR stage compared to





the other two stages, ice nucleation from marine aerosols becomes more efficient (Fig. 10d). The
mostly significantly invigorated postfrontal cloud cells by the marine INP effect (i.e., the
increase in both LWC and IWC and a large increase in cloud fraction) might also be related to
small scale thermodynamic changes through the feedback of microphysical changes over the first
two AR stages.

As for why increases of deep cloud occurrence and precipitation are less significant after

AR landfall compared to before AR landfall, first, we see the ice nucleation rates from dust INPs
is larger after AR landfall (the largest among the three stages; Fig. 10c), because of the increased
dust loading due to stronger transport (Fig. 10a). Stronger dust INP effects would limit the
marine INP effects since they compete for liquid drops. Second, the moisture increase after AR
landfall occurs in the lower atmosphere while the middle- and upper-level atmosphere are much
drier than before AR landfall (Fig. 6d), which favors more warm clouds and rain but is less
favorable to ice cloud development as indicated by the smallest ratio of snow precipitation (Fig.
7b). For more warm clouds/rain-dominated situations, the enhancement of ice formation would
have less influence.  Furthermore, in the drier condition aloft, more ice formation means less
efficient growth, thus the impacts on IWC/IWP and precipitation would be smaller. Cloud
dynamics (vertical velocity) is not changed much by the marine INP effect at all three stages,
indicating that the feedback from the increased latent heating resulting from enhanced deposition
and riming does not play an important role here, likely because this is not a convective
environment.
**4 Conclusion and discussion**

We have explored the effects of INPs from sea spray aerosols transported from the

Pacific Ocean on wintertime mixed-phase stratiform cloud properties and precipitation





associated with atmospheric river (AR) events. This is done by carrying out simulations at a
cloud-resolving scale (1 km) using WRF-Chem coupled with the spectral-bin microphysics
(SBM) scheme for an AR event observed during the 2015 Atmospheric Radiation Measurement
Cloud Aerosol Precipitation Experiment (ACAPEX). We have implemented the ice nucleation
parameterization for marine aerosols (McCluskey et al. 2018a) into SBM to account for the
marine INP effect. By comparing with available airborne and ground-based observations, we
show that considering the marine INP effect in the model improves the simulation of cloud phase
states (i.e., increased glaciation ratio) and precipitation.

Through enhancing ice and snow formation, marine INPs greatly enhance WBF and

riming processes, which convert liquid clouds to mixed-phase and ice clouds. There is a notable
shift in cloud occurrences with reduced shallow warm clouds (44%, 26%, and 7% for before and
after AR landfall and the post-AR stages, respectively) and increased mixed-phase (10%, 7%,
and 38% ) and/or deep cloud regimes (~ 22%, 8%, and 230%) because of the marine INP effect.
As a result, large increases in the ice water path (5 times or more), total condensate water path
(29% or more), and the ratio of snow precipitation (40 times or more) are seen. There is an
enhanced conversion of water from the vapor phase to the condensate phase so the water vapor is
generally reduced with the marine INP effect considered.

The significance of the above-described marine INP effects varies with the AR stages,

with a larger effect before AR landfall and post-AR than after AR landfall that has the dominant
precipitation. Note that the marine INP effects on cloud properties and snow precipitation are
still notable even at the stage after AR landfall. Although the total precipitation increases only by
4%, the drastic increase of snow precipitation and reduced rain precipitation at the surface have
an important implication for the regional water resources and flood risks since more snow



increases freshwater resources while less rain reduces flash flood risks. In addition, at this stage,
the marine INPs produce a notable spillover effect with a precipitation decrease (up to 30%) over
the windward slope of the mountains but precipitation (snow) over the lee side is doubled,
because more ice/snow formed over the windward side falls slower than rain and more easily
transported to the lee side.

Several reasons can be responsible for the smaller marine INP effects on cloud properties

(particularly reduction of shallow warm clouds and increased mixed-phase and deep clouds) and
snow precipitation after AR landfall compared with before AR landfall. First, the dust INP
effects are larger at this stage, which would limit the marine INP effect since they compete for
liquid drops. Second, after AR landfall, the moisture is heavily concentrated at the lower
atmosphere while the middle- and upper-level atmosphere are much drier than before AR
landfall. Therefore, the environment is more warm cloud and rain dominated, limiting the effects
of enhanced ice formation.  Furthermore, in the drier condition, more ice formation means less
efficient growth, thus the impacts on IWC/IWP and precipitation would be smaller.

The post AR stage has the largest response of the cloud regime shift and snow

precipitation among the three stages, because of the largest ice nucleation rates from the marine
aerosols. The larger ice nucleation rates compared with the other two stages are probably
because the lower cloud top heights (warmer than -25 ºC, i.e., shallower clouds) limit dust INP
nucleation, and with ~ 6 ºC colder temperatures below 8-km altitudes than the other two stages,
ice nucleation from the deep marine aerosol layer is more efficient.

This study suggests that the inclusion of marine INPs enhances orographic precipitation

mainly through more efficient growth (deposition and riming) of a larger number of ice particles
than liquid droplets, which is consistent with literature studies (Mühlbauer and Lohmann, 2009;



Fan et al., 2014, 2017; Xiao et al., 2015). The spillover effect by the increase of CCN has been
presented in several previous studies (e.g., Mühlbauer and Lohmann, 2008, 2009; Saleeby et al.,
2011, 2013; Carrio and Cotton, 2014; Letcher and Cotton, 2014). To our knowledge, this study is
the first to show the spillover effect associated with the INP effect. The prominent spillover
effect by the marine INP is different from Fan et al. (2014, 2017) that did not find such an effect
by dust INPs. There are a couple of factors that might be responsible for the difference. First,
marine INPs are mainly brought by ARs so the windward side gets INP first while dust INPs are
not associated with AR so there is no temporal sequence to have dust between the windward and
Lee sides. Second, the AR event is different with a different wind direction and speed, which
makes the transport of ice/snow to the lee side easier.

The marine INP effect revealed in this study is clearly emphasized due to the very low

dust INP concentrations for this particular situation and the deep marine aerosol layer during the
AR which allows notable ice nucleation at temperatures higher than -15 ºC. With high dust INPs,
the effects of marine INPs might not be as significant since they compete for supercooled liquid
drops. Although this is a single case study, the AR event and its evolution are representative.
Thus, the study suggests the importance of accounting for marine aerosols as INPs, in addition to
long-range transported mineral dusts, to simulate winter clouds and precipitation in the western
United States in regional and global climate models. We employ an empirical parameterization
for marine INPs developed from the data collected over the northern Atlantic Ocean and use sea
salt aerosols as a surrogate of marine organics, which might produce some uncertainties.
Nevertheless, the marine INP parameterization appears representative for this region based on
Levin et al. (2019). More observational data are needed in the western U.S. for (a) developing ice
nucleation parameterizations for potentially variable marine organics and (b) understanding



marine organics emission and chemical mechanisms and accurately simulating marine organics
in the model. As discussed earlier, the conversion of mass to number concentrations over each
aerosol bin might introduce some uncertainty to this study, which calls for model developments
of predicting the number concentration of each aerosol component.

**Data availability.**
The observational data can be accessed from the ARM data archive,
https://www.arm.gov/research/campaigns/amf2015apex.  The model simulation data will be
available through the NERSC data repository after the paper is accepted.

**Supplement.**
The supplement related to this article is available online.

**Author contributions.**
JF designed the study and model experiments. YL, JF, and PL performed numerical simulations
and analyses. JF and YL wrote the paper and other authors commented on it. LRL, PJD, LG, JF,
JT, YL, and JHJ contributed by either processing data including model input and observational
data or participating in the discussion of results.

**Competing interests.**
The authors declare that they have no conflict of interest.



**Acknowledgments.**
This research used resources of the PNNL Institutional Computing (PIC), and National Energy
Research Scientific Computing Center (NERSC). NERSC is a U.S. DOE Office of Science User
Facility operated under Contract No. DE-AC02-05CH11231. Funding for ACAPEX that
provides data collected on the G-1 aircraft was supported by the Atmospheric Radiation
Measurement (ARM) user facility, a U.S. Department of Energy (DOE) Office of Science user
facility managed by the Office of Biological and Environmental Research. The deployment of
the G-1 also involved the assistance of many PNNL/ARM field staff including M. Hubbell and
C. Eveland who flew the G-1 during ACAPEX. The authors acknowledge California Air
Resources Board for providing the 2015 emission inventory data and Dr. Allen White from
NOAA's Physical Sciences Laboratory for providing rainfall gauge data, and thank Alyssa
Matthews and Jingyu Wang at PNNL and Yuan Wang at JPL for technical/data discussion.

**Financial support.**
This study was supported by the Office of Science of U.S. Department of Energy Biological and
Environmental Research through the Regional and Global Model Analysis program area that
supports the Water Cycle and Climate Extremes Modeling (WACCEM) Science Focus Area at
PNNL and the DOE Early Career Research Program (project # 70071). PNNL is operated for the
U.S. Department of Energy (DOE) by Battelle Memorial Institute under contract DE-AC05-
76RL01830.





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





**Figures**

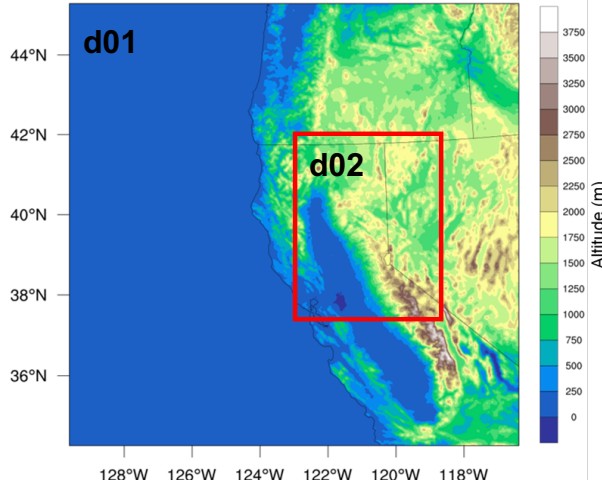


**Figure 1.** Two nested simulation domains: d01 and d02 centering over California. The color
shading denotes the terrain elevation.



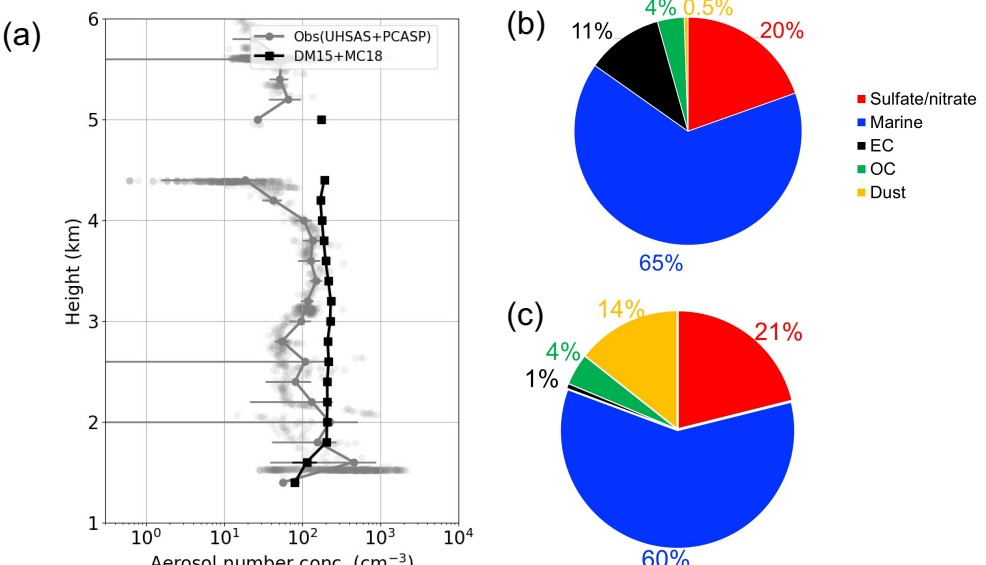


**Figure 2.** (a) Vertical distributions of aerosol number concentrations from aircraft observations

(Obs, grey) and DM15+MC18 (black) for particles with a dry diameter over a range of 0.067~3

μm, (b) mean fractional number contributions of aerosol classifications based on measurements

of single-particle mass spectra of aerosols and cloud particle residuals reported in Levin et al.

(2019), and (c) mean fractional mass contributions of aerosols in DM15+MC18 (number

concentration for each aerosol component is not predicted by WRF-Chem). The aerosol number

concentration from aircraft observations in (a) consists of both measurements from UHSAS and

PCASP. The modeled data in (a) and (c) are sampled along the aircraft route on 7 February 2015.

885

886



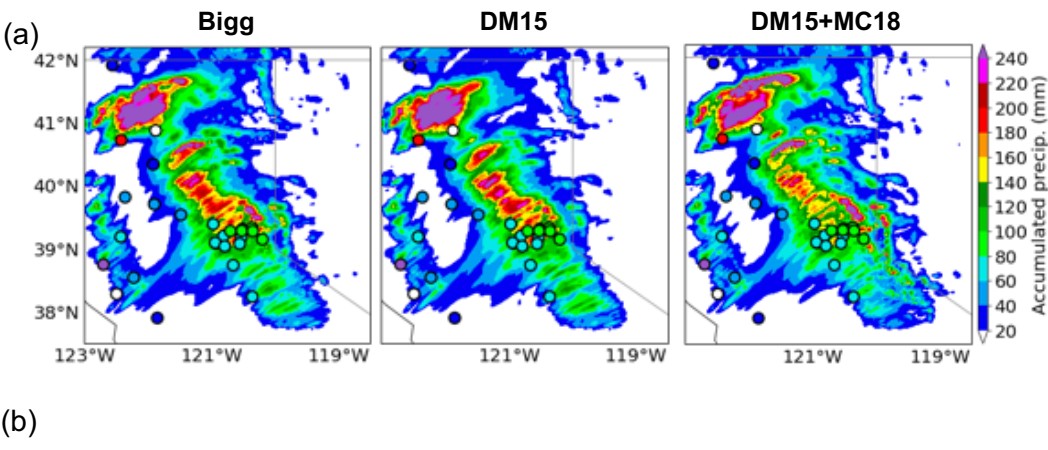

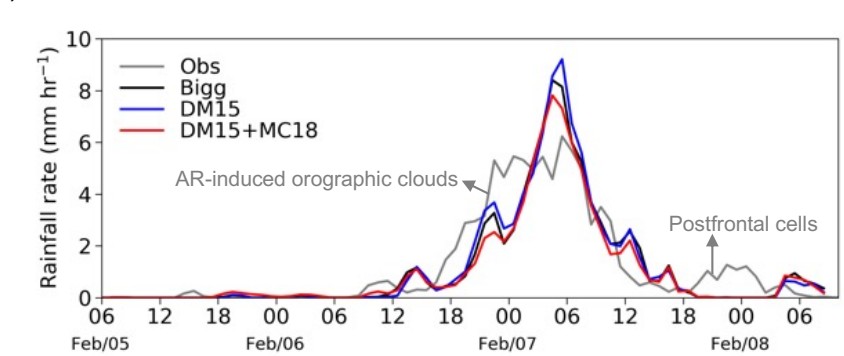

**Figure 3.** (a) Spatial distributions of accumulated precipitation during the AR event (06:00 UTC
5– 09:00 UTC 8 February). The color shading is for simulations and the circles denote the rain
gauge measurements provided by NOAA's Physical Sciences Laboratory. (b) Time series of
precipitation rates during the entire AR event for rain gauge observations (grey), and the
simulations of Bigg (black), DM15 (blue), and DM15+MC18 (red). The precipitation rates are
averaged over all the rain gauge sites shown in (a) for both observations and simulations. The
observed AR-induced orographic clouds and postfrontal cells are marked in (b).

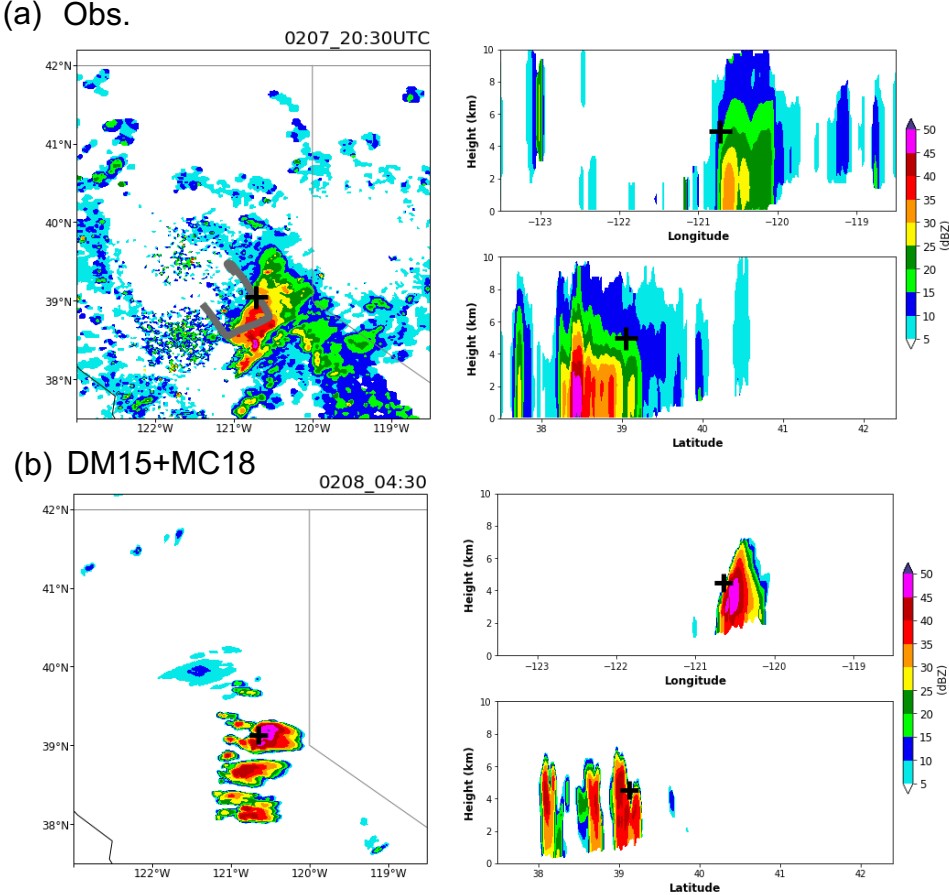

895

**Figure 4.** (a) Composite reflectivity of NEXRAD for the postfrontal clouds that the G-1 aircraft
sampled, (b) composite reflectivity from the simulation of DM15+MC18 for the postfrontal
clouds. The observation and simulation are compared at the peak reflectivity time which is 20:30
UTC 7 February for the observed clouds and 04:30 UTC 8 February for the simulated clouds.
The black crosses in the left two panels denote the positions where the longitude-height and
latitude-height cross-sections in the right panels are plotted. The grey line in the left panel of (a)
shows the flight track of the G-1 aircraft.



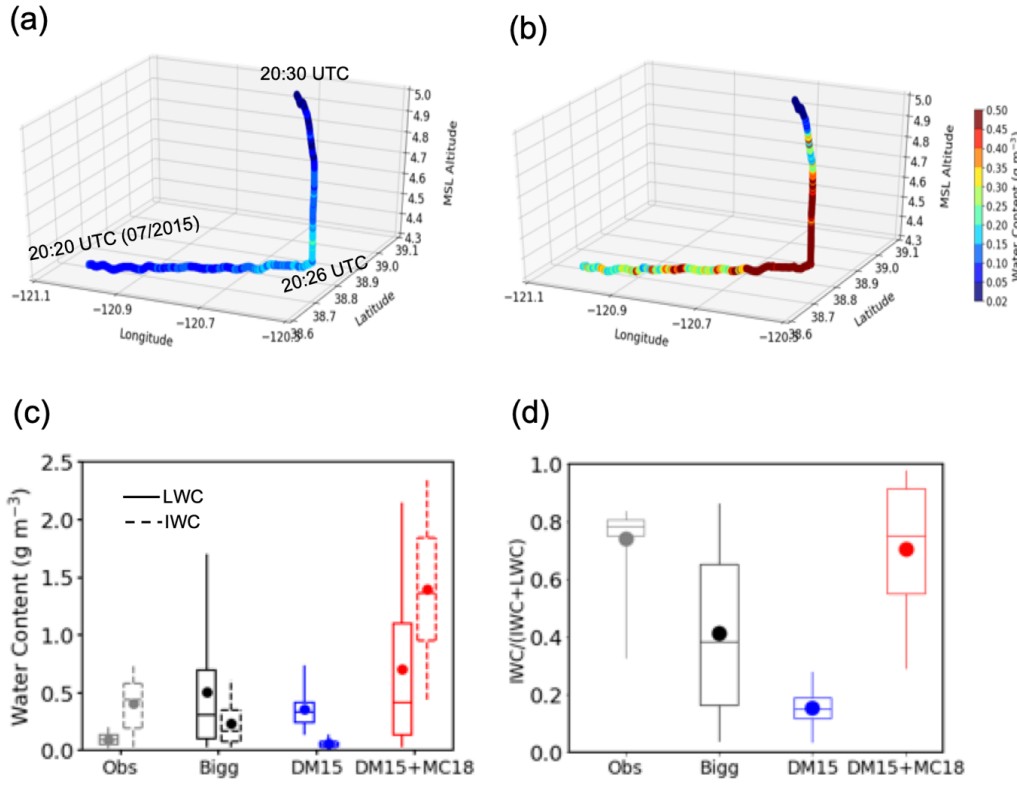


**Figure 5.** Comparisons of the simulations with aircraft observations. The observed (a) LWC and
(b) IWC along the flight track during 20:20 – 20:30 on 7 February when the aircraft flew through
the mixed-phase regime of the postfrontal clouds. (c) LWC (solid) and IWC (dashed) and (d) the
glaciation ratios of IWC/(IWC+LWC) from the aircraft measurements (Obs, grey) and
simulations of Bigg (black), DM15 (blue), and DM15+MC18 (red). The boxes show the 25th,
median (horizontal lines in the box), and 75th percentiles of the data. The upper and lower
whiskers show the 95[th] and 5[th] percentiles, respectively. The mean values are denoted by circles.

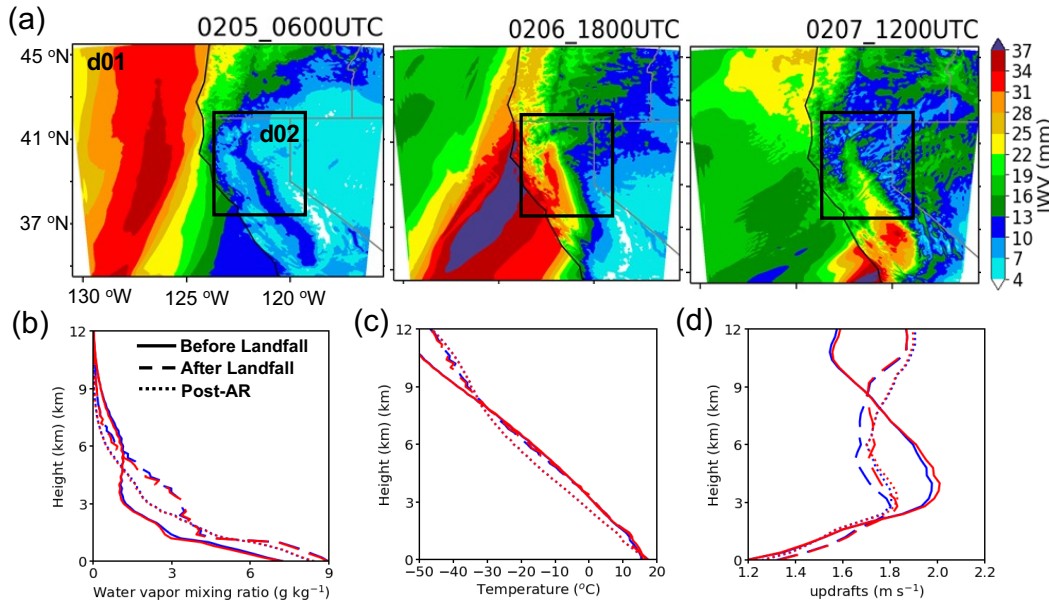


**Figure 6.** (a) Evolution of integrated water vapor (IWV) at 06:00 UTC 5 February (before AR

landfall), 18:00 UTC 6 February (after AR landfall), and 12:00 UTC 7 February (post-AR). The

black box (i.e., d02) in (a) is the domain of this study with the 5 lateral boundary grids excluded

for analysis at each side. (b-d) show the mean vertical profiles of (b) water vapor mixing ratio,

(c) temperature, and (d) updraft velocity at the three AR stages, i.e., before (solid lines) and after

(dashed lines) AR landfall and post-AR stages (dotted lines), for the simulations of DM15 (blue)

and DM15+MC18 (red). The water vapor mixing ratio and temperature are averaged for cloud-

free grids, and updraft velocity is averaged over the grids with a vertical velocity greater than 1

m s[-1].



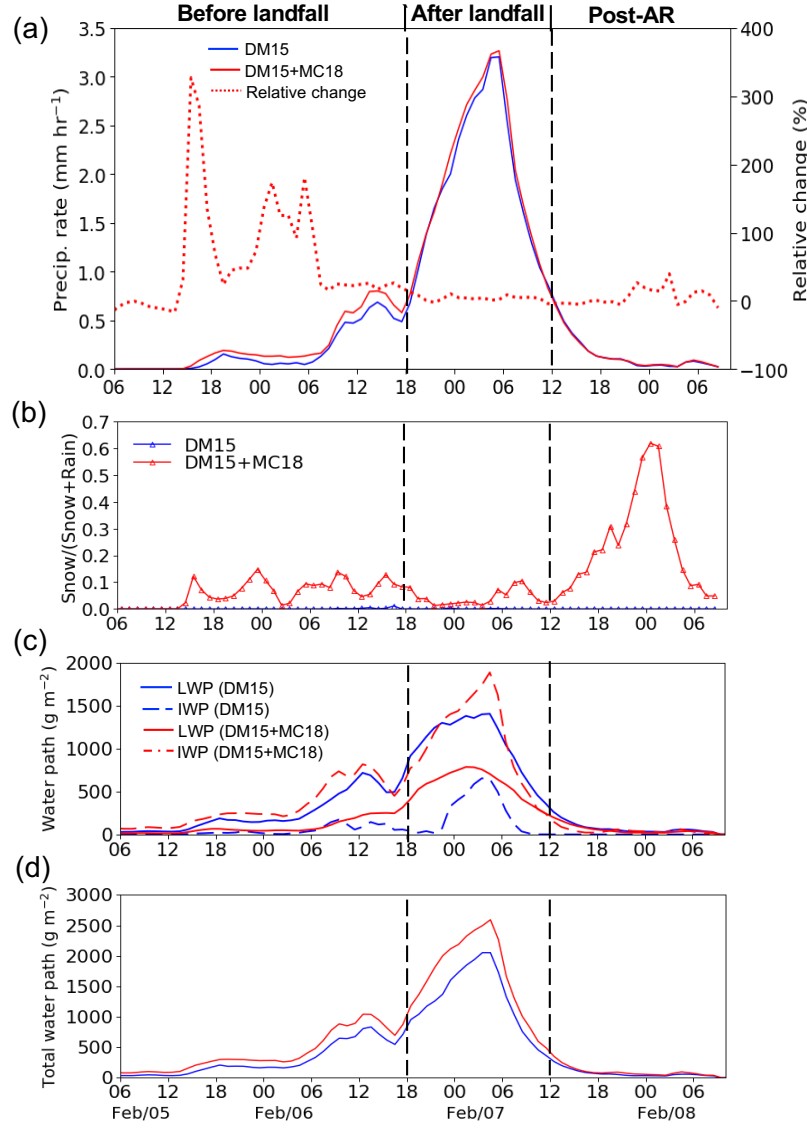

922

**Figure 7.** Time series of (a) precipitation rate (solid lines, left y-axis), (b) ratio of snow
precipitation (i.e., snow/(snow+rain) in mass mixing ratio) at the lowest model level, (c) LWP
(solid) and IWP (dashed) for DM15 (blue) and DM15+MC18 (red), and (d) total condensate
water path (TWP). The relative changes in precipitation rate from DM15 to DM15+MC18 are
shown in the red dotted line in (a) with values shown on the right y-axis. The vertical dashed
lines divide the three AR stages.



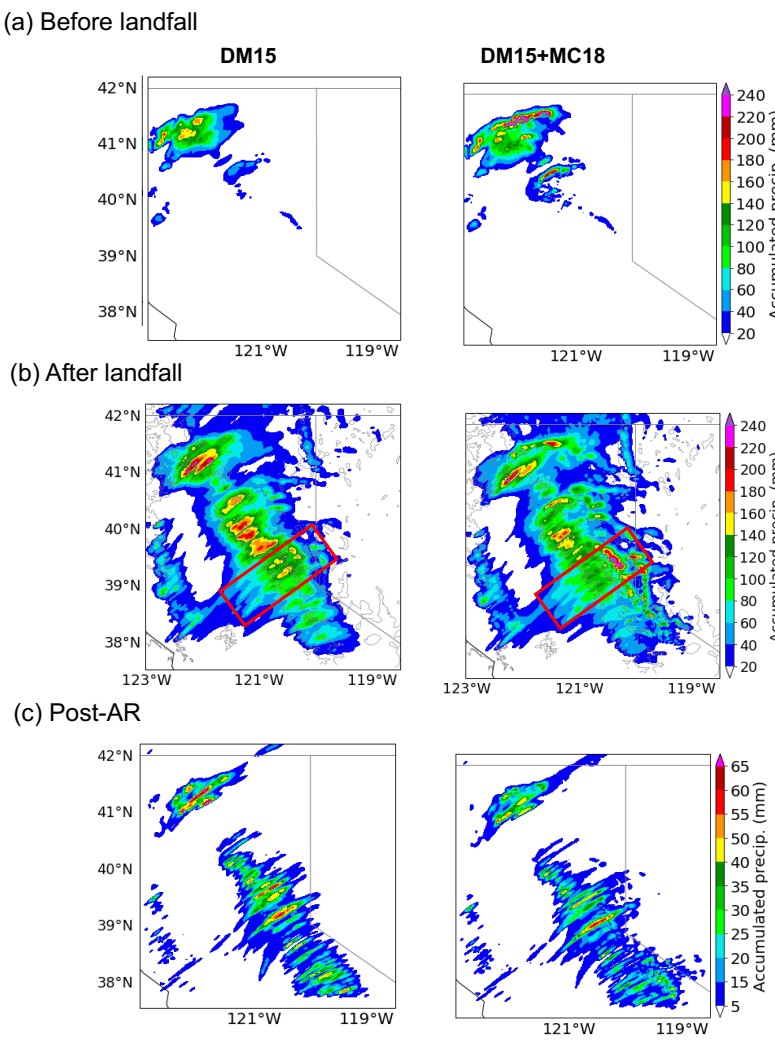

930

**Figure 8.** Spatial distribution of accumulated precipitation during the stages of (a) before AR

landfall, (b) after AR landfall, and (c) post-AR in DM15 (left) and DM15+MC18 (right). The

parallelograms marked in (b) denotes the area for the east-west cross-section analysis shown in

Figure 9.




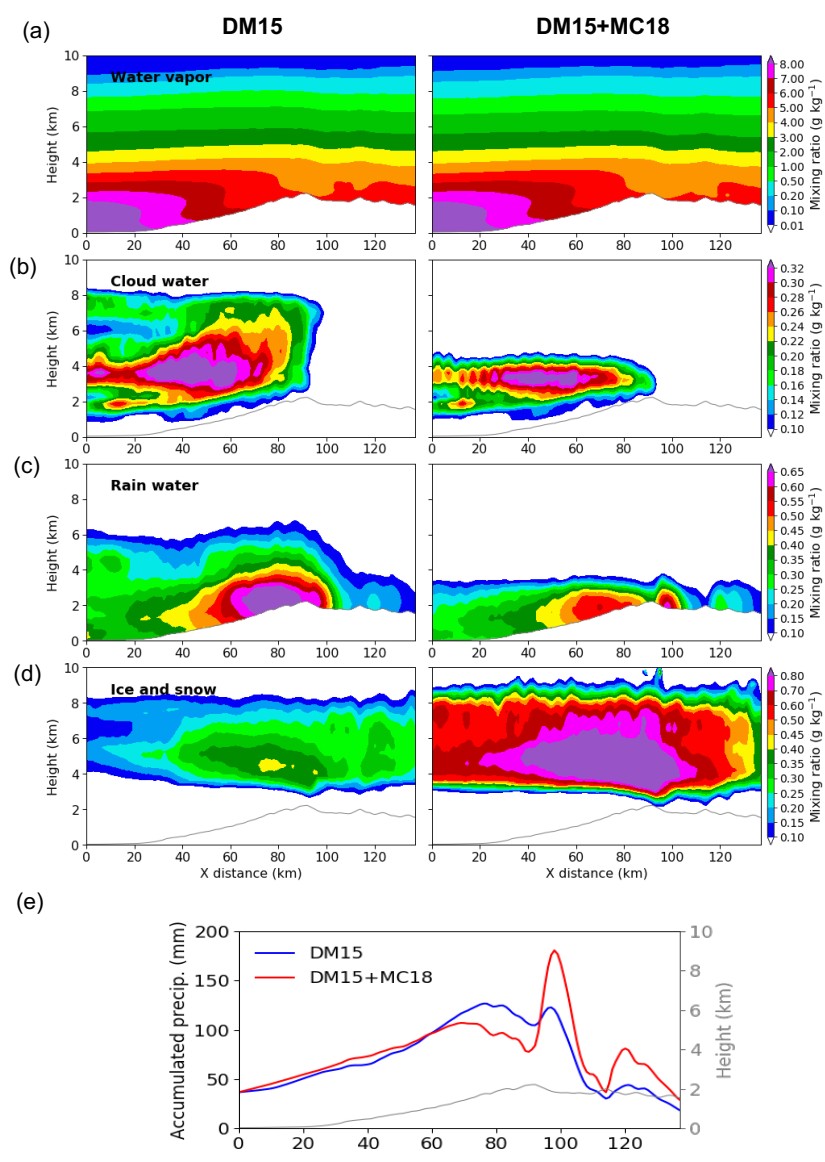



**Figure 9.** Mean mixing ratios of (a) water vapor, (b) cloud water, (c) rainwater, (d) ice + snow,
and (e) surface precipitation at the stage after AR landfall in DM15 and DM15+MC18. The
vertical cross-sections are averaged over the red boxes marked in Fig. 8b and the entire stage.




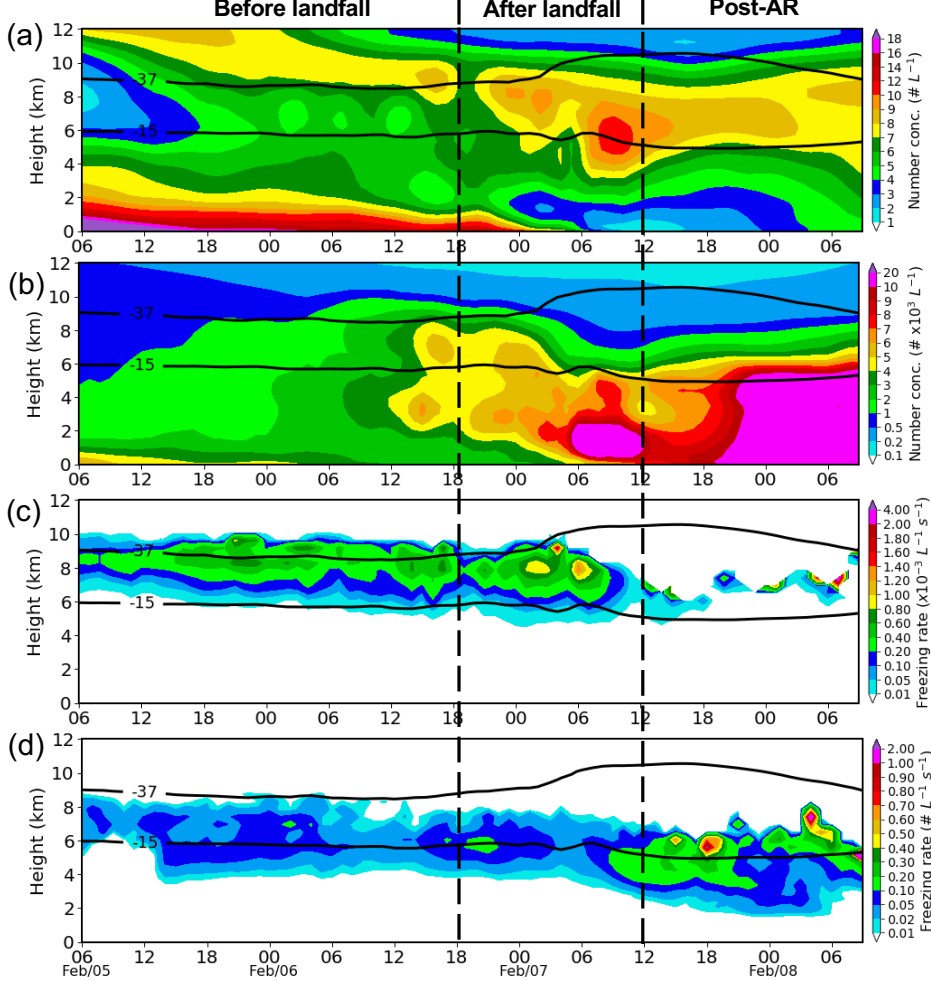


**Figure 10.** Time-height cross-sections of (a) dust particle (>0.5 μm) number concentration, (b) marine aerosol number concentration, (c) the freezing rate in DM15, and (d) the freezing rate in DM15+MC18. The number concentrations in (a) and (b) are derived from their corresponding mass mixing ratios under the clear-sky condition only. The freezing rates in (b) and (d) are the ice nucleation rates via immersion freezing at T > - 37 °C and the drop homogenous freezing rates at T < - 37 °C, and the values are for cloudy-points only. The black contour lines in each panel mark the temperature levels of -15 and -37 °C, representing the efficient immersion freezing temperature in DM15+MC18 and the homogeneous freezing temperature in the model, respectively.



954

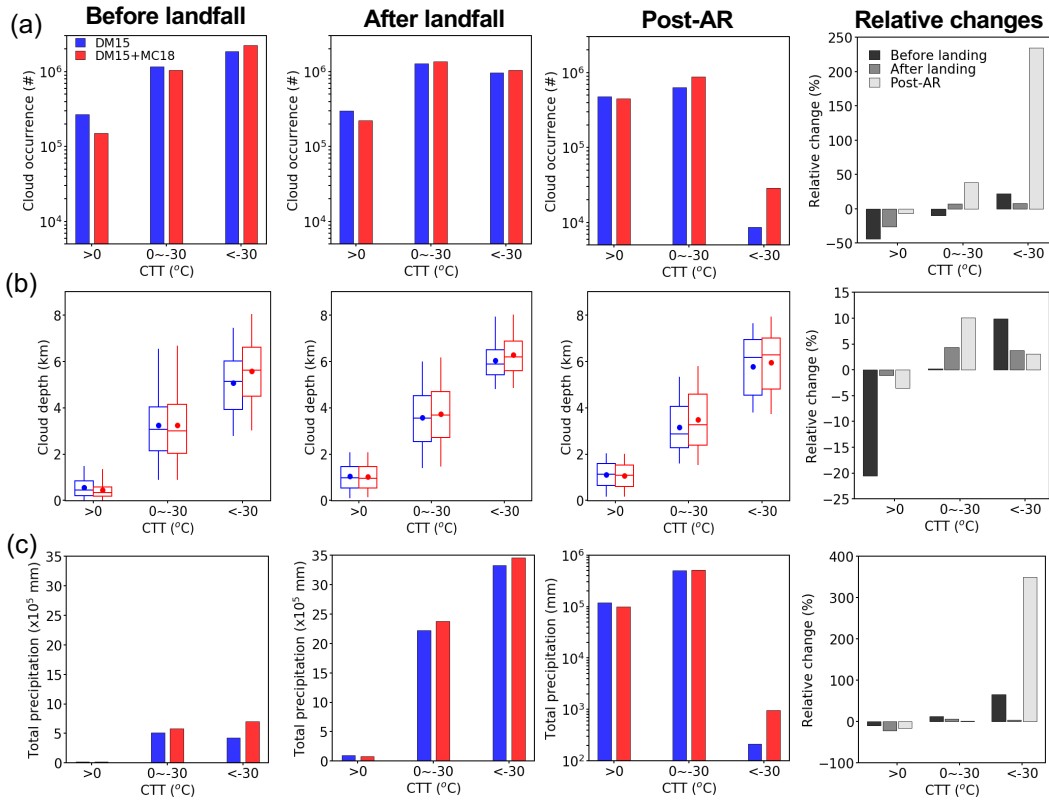

955

**Figure 11.** (a) cloud occurrences, (b) cloud depth, and (c) total precipitation for three cloud
regimes in DM15 (blue) and DM15+MC18 (red) at three AR stages from left to right: before AR
landfall, after AR landfall, post-AR. The last column shows the relative changes caused by the
marine INP effect, which are calculated as [(DM15+MC18) – DM15]/DM15*100%. Note that
the total precipitation at the post-AR stage uses a log scale for the y-axis. The box-whisker plots
follow the description in Figure 5c.



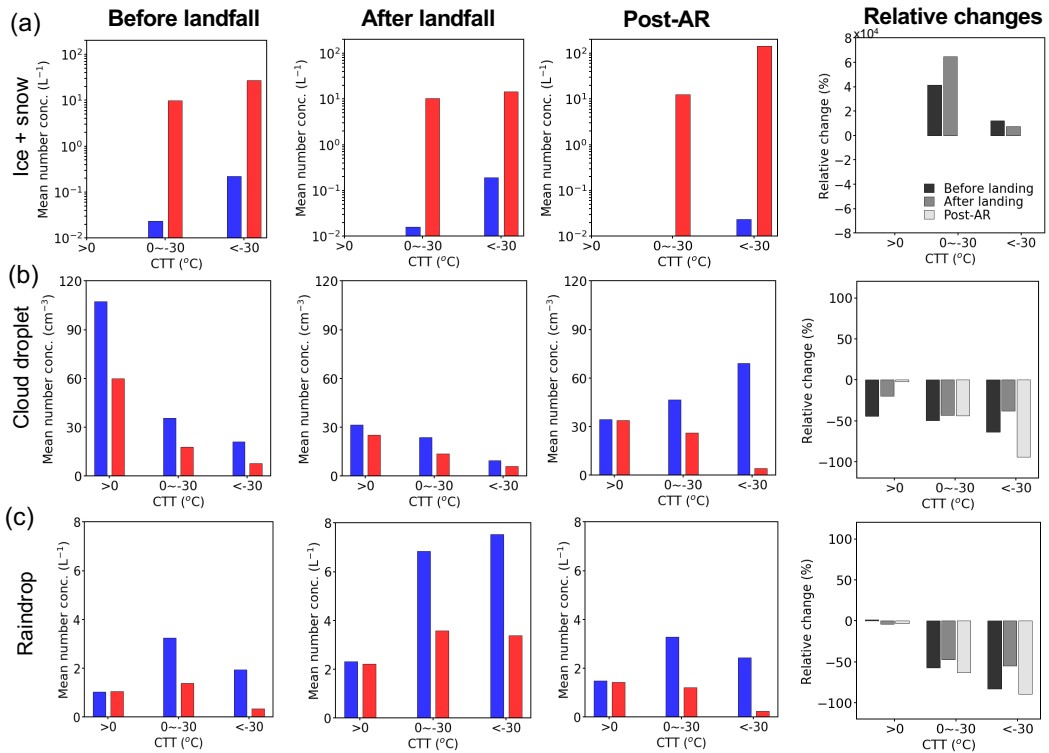

962

**Figure 12.** Hydrometeor number concentrations and their relative changes in three cloud regimes
in DM15 (blue) and DM15+MC18 (red) at the three AR stages for (a) ice particles (sum of ice
and snow), (b) cloud droplets, and (c) raindrops. The last column shows the relative changes
caused by the marine INP effect, which are calculated as [(DM15+MC18) –
DM15]/DM15*100%. Since ice particles are very limited at the post-AR stage in DM15, the
percentage changes of ice particles from DM15 to DM15+MC18 are huge numbers that are
omitted from the plots.

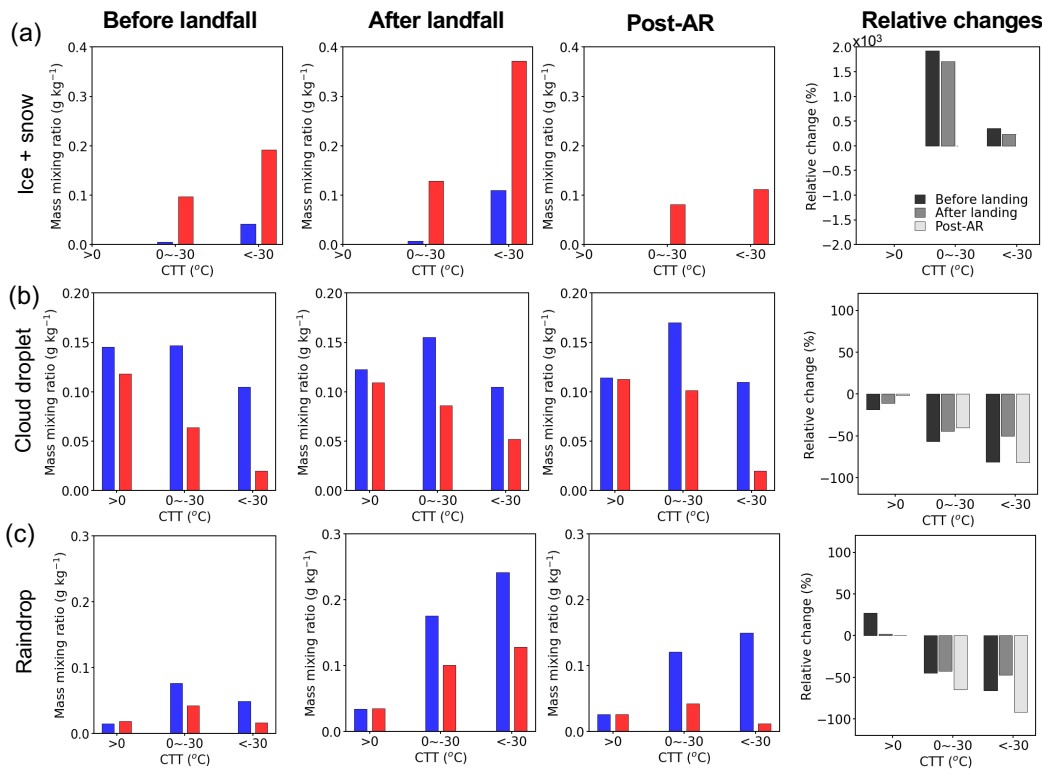


**Figure 13.** Same as Figure 12, except for the mass mixing ratios of (a) ice particles (sum of ice

and snow), (b) cloud droplets, and (c) raindrops.













**Table 1.** The changes in total precipitation, total condensate water path (TWP), liquid water path
(LWP), and ice water path (IWP), and cloud fractions (CF), net cloud radiative forcing (CRF) at
TOA from DM15 to DM15+MC18 (i.e., the marine INP effect), as well as the glaciation ratio,
i.e., IWC/(LWC+IWC), and the ratios of snow precipitation, i.e., snow/(rain+snow) in mass
mixing ratio at the lowest model level from DM15 to DM15+MC18, at the three AR stages. The
percentage changes are calculated following ((DM15+MC18)- DM15)/DM15*100.

| AR stages | | Before landfall | After landfall | Post-AR |
|---|---|---|---|---|
| Total precipitation | | 36% | 4% | -1% |
| TWP | | 45% | 29% | 35% |
| LWP | | -66% | -46% | -26% |
| IWP | | 8 times | 5 times | 440 times |
| CF | | 5% | 4% | 20% |
| Net CRF at TOA | | 15% | 13% | 10% |
| IWC/(LWC+IWC) | DM15 | 0.14 | 0.16 | 0.001 |
| | DM15+MC18 | 0.74 | 0.59 | 0.36 |
| Snow/(Rain+Snow) | DM15 | 0.002 | 0.001 | <0.001 |
| | DM15+MC18 | 0.085 | 0.042 | 0.131 |







**Table 2.** The domain-mean mass rates of deposition and riming in the mixed-phase and deep
cloud regimes in DM15 and DM15+MC18 at the three AR stages.

| AR stages | | Before landfall | | After landfall | | Post-AR | |
|---|---|---|---|---|---|---|---|
| | | Mixed-phase clouds | Deep clouds | Mixed-phase clouds | Deep clouds | Mixed-phase clouds | Deep clouds |
| Deposition ($mg\ kg^{-1}\ h^{-1}$) | DM15 | 44 | 171 | 81 | 388 | 7 | 8 |
| | DM15+MC18 | 846 | 780 | 1128 | 1397 | 781 | 1013 |
| Riming ($mg\ kg^{-1}\ h^{-1}$) | DM15 | 27 | 89 | 57 | 297 | 25 | 34 |
| | DM15+MC18 | 377 | 228 | 575 | 858 | 505 | 361 |

