# Peer review of "Modeling impacts of ice-nucleating particles from marine aerosols on mixed-phase"

_Atmospheric Chemistry and Physics, 2021_

## Referee Comment (RC2)

**Review for:**

**Impacts of ice-nucleating particles from marine aerosols on mixed-phase orographic clouds during 2015 ACAPEX field campaign**

*by Lin et al.*

This study examines the impacts of marine INPs on orographic clouds and precipitation associated with atmospheric rivers. For their investigations, they simulate an episode observed during the ACAPEX campaign with the WRF-Chem model, coupled with a spectral-bin microphysics scheme. They find that marine INPs have significant influence during the periods before and after the atmospheric river event, resulting in increased snow formation and precipitation.

This is a well-written paper and a very interesting study, as the role of marine INPs in the atmosphere remains poorly quantified. The study also includes some novel modeling aspects, such as the inclusion of a freezing parameterization for marine aerosols in WRF-Chem. For this reason, I recommend the manuscript for publication after some minor comments below have been addressed.

**Comments:**

– The paper includes a rather long introduction which provides brief information about the ACAPEX campaign and the examined case study. I would suggest making the introduction shorter and add two separate sub-sections about: (a) the field campaign, the utilized instruments and their respective uncertainties, and (b) a description of the examined event (meteorological and aerosol conditions). I think it would be very helpful for the reader to have a clear view of the episode's characteristics before reading section 3. Also information on instrumentation is scattered in the manuscript, while it would be better if this was gathered in a separate section, again before section 3.

– **lines 144-145:** *The SBM scheme is a fast version in which ice crystal and snow (aggregates) in the full version.* I don't understand the meaning of this sentence could you explain in more detail?

– **line 263:** I assume that cumulus parameterization is neglected in both domains. However it would be good to state that also in the paper.

– **line 296:** *shown in a later figure.* Please state the number of the figure

– **line 296-297:** *This is because dust is mainly from aerosol bins at larger sizes.* I guess this is something indicated by the measurements? If so, specify, and if possible provide information on the prevalent dust particle size range that was observed.

– **Line 313-316:** This is not very obvious to me as the three simulations look very

similar. Maybe it would be better if you could provide a mean precipitation value in the text for the region you are examining in these lines (and also be more specific about the exact location where these differences are observed)

– **Line 317-318:** When I first read about the spillover effect here, I was surprised that this is simply mentioned as a hypothesis with no further detailed investigations. Then I figured out that this would be further examined in another subsection. It worths mentioning here that this will be discussed in more detail in section 3.2

– **Line 319-320:** Again this difference is not very prominent at latitudes >40N. Either provide a mean estimate for the examined region or maybe show contourplots of the difference between the different runs

– **Line 333-334:** To solve this problem, many WRF studies conduct the simulations in segments (e.g. in 48-hour segments including a 24-hour spin-up after each initialization). Then they concatenate the outputs from the different segments. Consider adapting this method in your study

– **Figure 5:** While indeed the inclusion of MC18 parameterization substantially improves cloud fraction, the representation of total condensate is in worst agreement with observations. This is not mentioned in the text at all, while it would be useful to have a more quantified discussion on these discrepancies.

– **Line 393:** It is not very obvious to me how such large differences (>100%) in precipitation are estimated from Figure 7a, while precipitation rates are so close for the two runs

– **Figure 9:** I find very interesting that the vertical structure of liquid and ice is so different between the two simulations. Is it possible to evaluate which structure is closer to observations? Did the aircraft make some vertical profiling of cloud properties? In Figure 5a only a relative shallow LWC/IWC profile is presented

– **Line 463:** homogeneous freezing rates are mentioned in the caption of this figure but not in this text line.

– **Line 471:** while differences in nucleation rates at temperatures above the $-15^{\circ}$C isotherm are discussed, this is not the case for differences below $-37^{\circ}$C (which are also very prominent).

– **Table 1:** why some values are discussed in %, others in 'times' and other parameters are presented in absolute values? It would make more sense to use the same approach for all parameters

---

## Author Comment (AC1)

**Response to Reviewer #1:**

**Review for:**

**Impacts of ice-nucleating particles from marine aerosols on mixed-phase orographic clouds during 2015 ACAPEX field campaign**

*by Lin et al.*

This study examines the impacts of marine INPs on orographic clouds and precipitation associated with atmospheric rivers. For their investigations, they simulate an episode observed during the ACAPEX campaign with the WRF-Chem model, coupled with a spectral-bin microphysics scheme. They find that marine INPs have significant influence during the periods before and after the atmospheric river event, resulting in increased snow formation and precipitation.

This is a well-written paper and a very interesting study, as the role of marine INPs in the atmosphere remains poorly quantified. The study also includes some novel modeling aspects, such as the inclusion of a freezing parameterization for marine aerosols in WRF-Chem. For this reason, I recommend the manuscript for publication after some minor comments below have been addressed.

We thank the reviewer for your time and helpful comments. Our point-by-point response is enclosed.

**Comments:**

– The paper includes a rather long introduction which provides brief information about the ACAPEX campaign and the examined case study. I would suggest making the introduction shorter and add two separate sub-sections about: (a) the field campaign, the utilized instruments and their respective uncertainties, and (b) a description of the examined event (meteorological and aerosol conditions). I think it would be very helpful for the reader to have a clear view of the episode's characteristics before reading section 3. Also information on instrumentation is scattered in the manuscript, while it would be better if this was gathered in a separate section, again before section 3.

We have shortened the introduction, mainly on the INP description. We have added another section - section 3 for case description and measurements and their uncertainties (right before the Result section), since those details are not appropriate to put in the introduction section. This also leads to moving the original Fig. 6 up to Fig. 2 in the revised manuscript.

– **lines 144-145:** *The SBM scheme is a fast version in which ice crystal and snow (aggregates) in the full version*. I don't understand the meaning of this sentence could you explain in more detail?

The sentence has been clarified as "The SBM scheme is a fast version in which ice crystal and snow (aggregates) are represented with a single size distribution (low-density ice) with a

separation at 150 μm in radius, and graupel or hail is for high-density ice represented with an additional size distribution" (Line 137-138)

– **line 263:** I assume that cumulus parameterization is neglected in both domains. However it would be good to state that also in the paper.

We have added the sentence "Cumulus parameterization is not considered for the simulations over both domains" to the last sentence of the 2nd paragraph on Page 12.

– **line 296:** *shown in a later figure*. Please state the number of the figure

It is Figure 10. Since a few figures before this figure have not been cited yet, we are not allowed to cite that figure. Here the main purpose is to let readers now it will be further discussed with figures later.

– **line 296-297:** *This is because dust is mainly from aerosol bins at larger sizes.* I guess this is something indicated by the measurements? If so, specify, and if possible provide information on the prevalent dust particle size range that was observed.

Here the discussion is about simulated dust mass and number along the aircraft path. We meant to say the dust number concentration is dominated by small particles but it was said oppositely. Thanks for capturing the mistake. It is now corrected, and numbers are provided, i.e., "Although the simulated dust mass fraction is ~14%, the derived number concentration for sizes larger than 0.5 μm is very low (less than 0.02 cm$^{-3}$, shown in a later figure). This is because the dust number concentration is dominated by small particles (14.71 cm$^{-3}$ for the sizes smaller than 0.5 μm)". (Lines 344-346)

– **Line 313-316:** This is not very obvious to me as the three simulations look very similar. Maybe it would be better if you could provide a mean precipitation value in the text for the region you are examining in these lines (and also be more specific about the exact location where these differences are observed)

We have added white boxes in Fig. 4a to mark up the region we are examining. We also discussed mean precipitation over the white box area. The differences in precipitation rates between the simulations and observations were also plotted and shown in Fig. 4c (note we did not plot the differences with percentage because some very low values in denominator make huge values in percentage). All quantitative discussion has been added as "All three simulations predict a narrower but higher peak precipitation compared with the observed wider peak with lower values (Fig. 4b). However, the overestimation of the peak value by DM15+MC18 is lower than the other two (30% vs. 45% for DM15 and 58% for Bigg; Fig. 4b-c). The accumulated precipitation in the southern mountain range (the southern part of white boxes in Fig. 4a) is generally less than 100 mm in observations and less than 120 mm in DM15+MC18 but more than 140 mm in other two simulations. The mean precipitation over the white box accumulated over the AR period are 89, 128, 130, and 116 mm for observations, Bigg, DM15, and DM15+MC18, respectively. Again, although all three simulations overestimate the precipitation, DM15+MC18 simulates the lowest value and closer to observations" (Lines 358-367).

**– Line 317-318:** When I first read about the spillover effect here, I was surprised that this is simply mentioned as a hypothesis with no further detailed investigations. Then I figured out that this would be further examined in another subsection. It worths mentioning here that this will be discussed in more detail in section 3.2

As suggested, "This will be discussed in more detail in section 4.2" has been added (Line 370).

**– Line 319-320:** Again this difference is not very prominent at latitudes >40N. Either provide a mean estimate for the examined region or maybe show contourplots of the difference between the different runs

We calculated the accumulated precipitation during 06:00 UTC 5– 09:00 UTC 8 February averaged over the region with latitudes greater than 40° N and the values are 45, 42, and 48 mm for Bigg, DM15, and DM15+MC18, respectively.  The text was modified as "In the northern part of the domain (> 40º N), DM15+MC18 predicts more precipitation (i.e., 48 mm for the mean accumulated precipitation) than the other two simulations (i.e., 45 mm in Bigg and 42 mm in DM15" (Lines 371-373).

**– Line 333-334:** To solve this problem, many WRF studies conduct the simulations in segments (e.g. in 48-hour segments including a 24-hour spin-up after each initialization). Then they concatenate the outputs from the different segments. Consider adapting this method in your study

Thanks for suggestion. Indeed, we sometimes employed this approach. For this case, we did not adopt it because we wanted to simulate the entire AR event continuously without resetting the simulation in the middle of the event.

**– Figure 5:** While indeed the inclusion of MC18 parameterization substantially improves cloud fraction, the representation of total condensate is in worst agreement with observations. This is not mentioned in the text at all, while it would be useful to have a more quantified discussion on these discrepancies.

This was discussed (now Lines 403-407). We have modified it by adding the quantitative values which are shown in Figure. That is, "LWC is overestimated in all three simulations with DM15+MC18 of the largest overestimation (6 times higher than observation), while IWC is underestimated in Bigg and DM15 (nearly an order of magnitude lower in DM15 than observation) (Fig. 6c). DM15+MC18 predicts much higher IWC than the other two simulations, with an overestimation of IWP by ~3 times".

**– Line 393:** It is not very obvious to me how such large differences (>100%) in precipitation are estimated from Figure 7a, while precipitation rates are so close for the two runs

We calculated the relative change in percentage as [(DM15+MC18) – (DM15)] /(DM15)*100%. Since the precipitate rates in DM15 at some point before AR landfall are very small, the large increase from DM15 to DM15+MC18 is difficult to see. For example, at 16:00 Feb. 5, the percentage increase is 327% from 0.008 mm/hr in DM15 to 0.033 mm/hr in DM15+MC18.

**– Figure 9:** I find very interesting that the vertical structure of liquid and ice is so different between the two simulations. Is it possible to evaluate which structure is closer to observations? Did the aircraft make some vertical profiling of cloud properties? In Figure 5a only a relative shallow LWC/IWC profile is presented

The aircraft did not sample much in the vertical direction. Fig. 6 showed the flight path, and vertically it only spans over 4.3 -5.0 km, which is only 1-2 vertical layers in model.

**– Line 463:** homogeneous freezing rates are mentioned in the caption of this figure but not in this text line.

This sentence talks about the heterogenous freezing rates from the DeMott et al. (2015) and MC2018 parameterizations. It is not for the description of Figure 10 or the simulations of DM15 and DM15+MC18.

**– Line 471:** while differences in nucleation rates at temperatures above the $-15\,^{\circ}$C isotherm are discussed, this is not the case for differences below $-37\,^{\circ}$C (which are also very prominent).

We have added the discussion about the differences in the homogenous freezing rates, i.e., "Homogenous freezing (< -37 ºC; Fig. 10d vs. 10c) occurs less in DM15+MC18 because of a larger consumption of liquid drops and supersaturation in the heterogenous freezing regime. This is commonly seen in convective clouds (e.g., Zhao et al. 2019)." (Lines 510-512)

**– Table 1:** why some values are discussed in %, others in 'times' and other parameters are presented in absolute values? It would make more sense to use the same approach for all parameters

We use % in general in this table. For IWP, because the increase is so large, for example, it will be 44000% for post-AR if percentage is used, we use "times" which is more straightforward to readers. For the glaciation ratio and snow precipitation ratio, it is more physically meaningful to show the ratio (actual value) instead of percentage change.

---

## Author Comment (AC2)

**Response to Reviewer #2:**

Review of:

"Impacts of ice-nucleating particles from marine aerosols on mixed-phase orographic clouds during 2015 ACAPEX field campaign"

Authors: Lin et al.

Recommend major revisions.

General comment:

Overall, I find the paper and topic interesting and relevant to the microphysics modeling community. The potential impacts of marine sea-salt particles as INP is particularly interesting in, that historically, they have not been considered efficient INP. So, exploring their impact within a controlled microphysics modeling environment is quite important. As you will see in my specific comments below, I think the authors need to give a fairer assessment of the INP impacts on modeled fields and not attempt to overemphasize effects that appear to be rather minor when viewing the figures. Please show and explain the results in a balanced manner.

We thank the reviewer for your time and helpful comments. Our point-by-point response is enclosed. We would like to point out that this is about the potential impact of sea-spray aerosols, not sea salt aerosols. Sea spray particles that can contain sea salt, but also organics importantly. It is just in the model we used sea salt as a surrogate for sea spray particles given that most marine organic aerosols exist with coating on the surface of sea salt particles in the size range that dominates surface area.

Specific comments:

1.Abstract line 36-37: What is the difference between "post-AR" and "after AR"?

"After AR landfall" (these three words need be read together) is a stage after AR made landfall with large increase of IVT and precipitation. "Post-AR" begins from the point when AR started to retreat. The different AR stages are defined in Section 3.

2.Introduction line 44: Please be more specific regarding AR impacts on the "western" U.S., specifically when you state that it accounts for 30-50% of the precipitation. By western, do you mean Pacific Coast states?

We meant California. The sentence has been revised as "On a long-term average, AR storms contribute to 20–50% of California's precipitation totals (Dettinger et al., 2011)" (Lines 44-45).

3.Section 2: Will you please provide the hydrometeor fall speed power law coefficients used in this version of HUCM-SBM for each ice species? I have found the power law coefficients to be quite important in such studies and would like to know what was used in this study.

The fall speed power law relationships are shown in a previous study https://doi.org/10.1175/MWR-D-16-0385.1 (see Table 1 and Fig. 1 of this study). We have added a sentence to cite this study, i.e., "The fall speed power law relationships for ice/snow and graupel are depicted in Xue et al. (2017)" (lines 140-141)

4. Line 260: The 40% reduction in aerosol number is quite a lot. How much closer to observations do you get with use of the CARB2015 dataset over NEI2011?

We have calculated the mean number concentrations of aerosols below 2 km from observations and simulations using CARB2015 and NEI2015 and added the discussion, i.e., "The aerosol concentration averaged over 1-2 km altitudes is about 160 cm$^{-3}$ with CARB2015 and 317 cm$^{-3}$ with NEI2015, which is 26% lower and 47% higher than aircraft observations (215 cm$^{-3}$), respectively. Thus, the simulated aerosol concentrations with CARB2015 are in better agreement with observations" (Lines 251-254).

5. Section 2.2: How do you get realistic dust transport over the Pacific into California in this scenario with limited spin-up time?

Our chemical and aerosol initial and boundary conditions are from the global WRF-Chem simulations, not from the original WRF-Chem setup which requires a week to spin up. That means the chemistry and aerosol fields used for our model simulations already have steady-state values. Therefore, 2 days spin-up for our simulations are good enough. Since our model simulations overestimates aerosol concentrations in general, there is no indication of insufficient spin-up time.

6. Line 297: Why is only larger dust present? Wouldn't the larger dust tend to settle out before the smaller dust particles?

We meant to say the dust number concentration is dominated by small particles but it was said oppositely. Thanks for capturing the mistake. It is now corrected, and numbers are provided, i.e., "Although the simulated dust mass fraction is ~14%, the derived number concentration for sizes larger than 0.5 μm is very low (less than 0.02 cm$^{-3}$, shown in a later figure). This is because the dust number concentration is dominated by small particles (14.71 cm$^{-3}$ for the sizes smaller than 0.5 μm)" (Lines 344-346).

7. Lines 316-323: This discussion on accumulated precipitation hinges on small changes seen in figure 3. It is difficult to see the changes being discussed in this manner. Perhaps figure 3 should include difference plots so that we can more readily see the spillover effects being discussed. As figure 3 is currently presented, all the simulations look very similar with only minor differences in the details as one would expect when changing a parameterization.

For the spatial distribution plots, difference plots have a problem to clearly show the comparison with the observations at the stations. We have used white boxes to mark up the exact location where these differences are observed (Figure 4a) and calculated mean precipitation for the white box area. Also, for the time series plot, we have added a panel for the differences between the simulation and observations (Fig. 4c). All quantitative discussion has been added as "All three

simulations predict a narrower but higher peak precipitation compared with the observed wider peak with lower values (Fig. 4b). However, the overestimation of the peak value by DM15+MC18 is lower than the other two (30% vs. 45% for DM15 and 58% for Bigg; Fig. 4b-c). The accumulated precipitation in the southern mountain range (the south part of white boxes in Fig. 4a) is generally less than 100 mm in observations and less than 120 mm in DM15+MC18 but more than 140 mm in other two simulations. The mean precipitation over the white box accumulated over the AR period are 89, 128, 130, and 116 mm for observations, Bigg, DM15, and DM15+MC18, respectively. Again, although all three simulations overestimate the precipitation, DM15+MC18 simulates the lowest value and closer to observations" (Lines 358-367).

8.Lines 333-334: While long time integration could be impacting cell formation, there are many other model artifacts that could be hindering better prediction compared to the obs. Since you are using this statement to justify moving forward with the analysis, there needs to be better justification or explanation for why the predicted cells differ from the observations. We need to be convinced that the simulations are trustworthy.

It will need quite a lot of effort to figure out factors leading to the model biases to provide more specifics here, which would also be a large distraction of the focus. We agree that many other things in model could affect the biases, and we have revised the sentence to "The deviations of the simulation from observations for the postfrontal clouds could be because of various reasons such as (a) the long-time model integration time (the 4th day after model initiation) and (b) the spatial mismatch of simulated and observed clouds since those postfrontal clouds are small" (Lines 385-387).

9.Lines 353-363: In this section the authors seems to be focused on the improvement to the glaciation ratio in the simulation with sea salt INP while downplaying the large overestimation in water content in figure 5c. Should the simulation with MC18 be considered "better" than the others?

We have added text to discuss the large overestimation of LWC and IWC by MC18 because the post-frontal clouds are invigorated a lot (see Lines 403-407). We specifically state that MC18 is only better in simulating the cloud phase states. Since all simulations do not predict the post-frontal clouds well, we would recommend focus on how significant those clouds are changed by marine INPs instead of which simulation is better.

10.Lines 393: The comment on the 330% increase in precipitation is very misleading here. From figure 7a, it can be seen that the 330% increase occurs from a VERY small absolute increase in precipitation at a time when precipitation rate is very small. These sorts of statements regarding the analysis that over emphasizes a small impact should be clarified or not included in the discussion. The overall changes in precipitation rate due to sea salt INP is quite small as seen in figure 7a, with at most 0.1 mm/hr change.

Yes, we agree. We have changed to "over 50% in general" in Lines 431-432 and we have emphasized small precipitation before AR landfall by adding a sentence "Note the precipitation

is very small at some point before AR landfall so the large increases might not mean that much" (Lines 435-436).

11. Lines 423-425: While the rain vs snow argument is valid for hydrologic reasons, you have already shown that the accumulated precipitation differences are very small between simulations. So, does this really matter?

We would say the ratio of rain vs. snow matters more to people and society than the accumulated precipitation. Also, the spatial distribution of the precipitation matters more than the accumulated precipitation over a large region to our stakeholders.

12. In general, the discussion of the spillover effect is quite interesting and perhaps should be highlighted rather than over-emphasizing minor changes in precipitation rate. Further, I find figure 9, and the discussion on the glaciation of the cloud, the most fascinating part of this story thus far. It is my opinion that these features should be emphasized earlier in the paper and place it in the context of figure 5c. Does the MC18 simulation produce too much overall condensate while better predicting the relative proportions of water to ice?

The spillover effect was brought up earlier when discussing Fig. 4a and more detailed discussion is provided later with two figures (Figures 8 and 9). So it is well highlighted. Because section 4.1 focuses on comparisons with observations and the discussion about marine INP impacts starts from Figure 7, those features are already discussed in the earliest places their appear and the reasons leading to such features are elaborated in the section when marine INP effects are focused. The same for the overestimations of condensate content by MC18, which is described in Figure 5c but further discussed in section 4.3. We also added more pointers/hinters for readers to connect those discussions.

 "Does the MC18 simulation produce too much overall condensate while better predicting the relative proportions of water to ice?", Yes, and it is discussed with Fig. 6 (original Fig. 5) in section 4.1.

13. Lines 528-532: Here you are stating that competition for ice nucleation between dust and marine INP explains the differences in deep cloud occurrence and precipitation, yet you stated earlier that the ice nucleation for dust and marine INP occurs in two different temperature regimes and thus different vertical locations in the cloud. So, how do they compete in this scenario if they are activated in different locations?

In fact, the much higher ice nucleation rate from marine INPs in this case is only because of the large amount of marine INPs, not because their nucleation temperature regimes are different. Sorry about the misleading sentence (we forgot to change it to be consistent with the discussion of Figure 10 on Page 23). Now throughout the paper, we have made a consistent argument.

Here we have removed the argument. After we replotted Figure 10 cand d with the same log scale, we did not see a clear competition between dust and sea-spray nucleation.

14.Lines 538-542: Here you state that cloud dynamics (vertical velocity) is not changed much. Yet in line 524 you discuss invigoration of postfrontal cloud cells. This appear contradictory. Please clarify and discuss how invigoration works in this scenario. You mention the term invigoration several times but have not explained the process.

Here "invigoration" means increased condensates and cloud fraction, not stronger convection, which was already defined with "The mostly significantly invigorated postfrontal cloud cells by the marine INP effect (i.e., the increase in both LWC and IWC and a large increase in cloud fraction) might…" (now Lines 564-566). We have avoided using the term "invigoration" in other places in the revised manuscript. There was a paragraph in Section 4.3 (now the last paragraph P. 25) devoted to discussing why the postfrontal clouds can be invigorated. i.e., "As discussed earlier, the largest ice nucleation rates from marine aerosols at the post-AR stage explain the largest marine INP effects among the three stages. The factors contributing to the larger ice nucleation rates include the increased abundance of marine aerosols compared to the previous two stages (Fig. 10b). In addition, with the ~ 6 °C colder temperatures below 8-km altitudes during the post-AR stage compared to the other two stages, ice nucleation from marine aerosols becomes more efficient (Fig. 10d). The most significantly invigorated postfrontal cloud cells by the marine INP effect (i.e., the increase in both LWC and IWC and a large increase in cloud fraction) might also be related to small scale thermodynamic changes through the feedback of microphysical changes over the first two AR stages".

15.Conclusion: The papers first referenced in the conclusion should be discussed earlier in the manuscript.

That is not necessary. Depending on how you summarize. We think our organization of results and summary makes sense.

Figure 2: Please show panel A on a linear scale rather than log scale. The log scale somewhat minimizes the larger differences in aerosol number between the obs and model. Near 2.8km altitude, I estimate the aerosol concentrations to be 40/cm3 (obs) and 200/cm3 (model).

Aerosol concentrations vary by a few magnitudes in the vertical profile so it is usually plotted with log scale. We have provided more quantitative discussions now, i.e., "The simulation overestimates the total aerosol number concentrations by ~ 2-times averaged over the altitudes of 2.2-3.2 km. At 2.8 km, the difference between the simulation (219 cm$^{-3}$) and observations (55 cm$^{-3}$) is about 4 times" (Lines 334-336).

Figure 10: Why is there no homogeneous ice nucleation in panel D? Is this a contouring issue since the values in panel D are much larger than panel C? Perhaps you could plot panels C and D on a common log scale so that we can see the comparable differences. Also, the figure caption should say "The freezing rates in (c) and (d) ...."

Yes, it is because values in Panel d were plotted in three magnitude higher values. We have replotted c and d with the same log scale (also replotted a and b with the same log scale). The typo in the figure cation is also corrected. Thanks for the suggestion.

---

## Author Response (AR2)

Second Review of:
"Impacts of ice-nucleating particles from marine aerosols on mixed-phase orographic clouds during 2015 ACAPEX field campaign"

General comment:

While some aspects of the manuscript have been improved, the authors need to better address my past comments that have been restated below. There are several places where the impacts of marine INP make the simulation worse, yet the authors gloss over these differences. Further, there are places where the marine INP simulation is comparably indifferent with respect to the "control" simulations, and these effects are exaggerated. The most dominant marine INP impact appears in the spillover of precipitation and the glaciation ratio. The authors need to give a more fair assessment of the simulations and the impacts of marine INP. I feel that it is important not to overstate tiny changes and also not to understate situations in which the marine INP cause more disagreement with the observations. A fair and balanced objective assessment is needed at all times.

We thank the reviewer for the further comments. We fully agree with fair assessments of the simulations and the impacts of marine INP and we have attempted to achieve that by examining every conclusion based on the data. From your specific comments below, we would like to bring up a few remarks about the limitations of the observations and evaluation and how marine INP effects are examined in our study so that we are on the same page:

- The poor performance of the model occurs only at the post-AR stage when cloud cells are small and very challenging to simulate, but the post-AR stage is not our focus as we noted in the paper. Our conclusions about marine INP effects are not derived from this stage. This is also one of the reasons we examined marine INP effects by separating AR into different stages.
- The evaluation of the clouds at the post-AR stage with the aircraft data is only for a single cell, which might not be generalized to the clouds in other locations of the domain. Therefore, we limit the discussion of the post-AR stage to avoid distraction from the main focus of the study. To address your concerns, in the revised version we removed the conclusion that adding marine INPs improves cloud simulation of phase state from the abstract, and we have added discussion of the implication of the overestimated supercooled liquid to the modeled marine INP effects.
- Our marine INP effects are derived from the model sensitivity tests. In other words, the effects are for the modeled clouds only. We do not have any claim that it is the observed marine INP effects. We do not have the observations to evaluate if the modeled marine INP effects are similar to the observed effects or not. This has been emphasized in the last section in the revised manuscript and we have also changed the title of the paper to "Modeling impacts of ice-nucleating particles from marine aerosols on mixed-phase orographic clouds during 2015 ACAPEX field campaign".

Please see our embedded point-by-point responses below.

Specific comments:

1. Figure 3: I stated in my first review that your wide logscale for aerosol profile comparisons is visually minimizing the differences between the observed aerosol concentrations and the simulated concentrations. You have now stated in the text that there's a 4x higher value for simulated aerosol concentration at 2.8km. However, you do not discuss the implications involved in running simulations with much higher aerosol numbers than observed. This will inherently over emphasize the aerosol INP effects compared to the observations.

- We thank the reviewer for emphasizing the need to discuss the model limitations and implications for modeling INP effects. First, we would argue in aerosol simulations, the number concentrations are difficult to simulate since major processes influencing aerosol number are not well understood such as aerosol nucleation. Four times higher is not something surprising. Second, we want to note that the 4 times difference is for the total aerosol concentration which is not used in our model parameterization for the effects of dust and marine INPs. The number concentrations of dust and marine SSA used in model parameterization are not predicted so they are derived from mass concentrations but there are no mass observations for a single aerosol component. Therefore, we cannot infer that marine INP effects are overestimated based on the 4 times difference in total aerosol concentration. Furthermore, this evaluation is only from the aircraft measurements at the post-AR stage, which is not our focus of the study. We do not know how well the aerosols are simulated before AR and after AR landfall. The lack of relevant observations for robust evaluation of model simulation was discussed in the last section, and we further revised it in this version and now it reads as "More observational data particularly on the extended spatial and temporal coverage are needed in the western U.S. for (a) evaluating model simulations more robustly, (b) developing ice nucleation parameterizations for potentially variable marine organics and (c) understanding marine organics emission and chemical mechanisms and accurately simulating marine organics in the model" (Line 662-664).

2. Figure 4 discussion and Lines 367-368: In my previous review I requested the inclusion of difference plots of accumulated precipitation so that we can better visualize how the precipitation changed, particularly over the white box area. Without a difference plot it is nearly impossible to discern the differences as pictured. Please provide difference plots that can be added to figure 4. You could plot one as (DM15 - Bigg) and (DM15+MC18 – Bigg). These 2 new panels would help us see the differences more clearly.

- In the previous round, we hesitated to add more panels since the results were clear to us. Now we have added the difference plots between Bigg and DM15 and between DM15+MC18 and DM15 as the second row (using DM15 as a reference to be consistent with the marine INP effects that we focused on). The conclusions are the same. The corresponding description of the plots includes "The simulated precipitation between Bigg and DM15 is very similar except for more precipitation in Bigg in the northern part of the domain (Fig. 4a-b)" (Line 370-371) and "There is a clear spillover effect caused by marine INPs (Fig. 4a-b, right). That is, with marine INPs considered in DM15+MC18, there is a notable decrease in accumulated precipitation (~ 30-50 mm) on the windward side but a large increase (~ 50-70 cm) on the lee side (Figure 4b, right)" (Line 373-376).

3. Line 388: It seems here you are simply disregarding the differences in cell heights and just moving forward? The cell depth is quite important given the need for cold temperatures for ice nucleation and the necessary depth for ice growth. You need to discuss the limitations in interpreting results if your simulation cannot reproduce the depths of the cells.

- Since we have noted that the post-AR cell produced negligible precipitation and it is not the focus of the study (Line 440-442, and 500-502), we indeed do not want to use a lot of text on this to distract our focus. Also, this evaluation is just for one single post-AR cell after 3-4 days into the simulation. We do not know if it can be generalized to all postfrontal clouds. Therefore, besides emphasizing the modeled INP effects might not be the same as what occurred in reality, we do not have a direction for discussing what this means to marine INP effect but we have tried to add discussion about overestimated supercooled LWC. We have added the emphasis and discussion in two places: section 4.3 "The effects of marine INPs on the postfrontal clouds might differ from the reality since based on very limited measurement data, the model seems not be able to capture those clouds well. The overestimated supercooled LWC can allow for more riming growth which may lead to a larger sensitivity to marine INPs." (Line 549-552), and the conclusion section "Since our model may not simulate clouds well at the post AR stage based on very limited measurement data, we emphasize that the large responses to marine INPs simulated at this stage might not reflect the effect in reality" (Line 633-635)
- Note the we can not say if the cell height is reproduced or not. Based on the reflectivity shown in Fig. 5 which speaks about precipitating particles, the modeled cell has a shallow precipitating core, but we do not know if the cloud top height is lower than observations or not. Note sure where the reviewer got information about the cell height.

4. Lines 413-414: You state that inclusion of marine INP improves the simulations of cloud phase states. As I discussed in my past review, the glaciation ratio is better, but the simulated amount of LWC and IWC, which is critical for accumulated precipitation, is the worst of the simulations. So, I disagree that the DM15+MC18 simulation is better. While it's crucial for a model to simulate both the total condensate and the water/ice ratio, I would argue that getting the total condensate amount is most important when the bottom line is the improvement of predicting accumulated precipitation and its spatial distribution.

- We acknowledge the reviewer's point of providing a fair assessment of model skill. Therefore our statement only emphasizes the glaciation ratio, which is improved drastically from 0.2 in DM15 to 0.70 in DM15+MC18; the observed value of 0.74 so the sentence "The inclusion of the marine INP effect improves the simulation of cloud phase states via enhancing heterogeneous ice formation through immersion freezing" is accurate. About the overestimation of LWC and IWC, this was clearly described in the text right above the statement regarding glaciation ratio. In the current version, we have also added discussion about the implication of high LWC in marine INP effect in section 4.3 (see our response to #3).

5. Lines 435-436: Your sentence in response to my past comment on figure 7a states: "Note that precipitation is very small at some point before AR landfall, so the large increases might not mean that much." This response is not adequate and appears a bit out of place here. I

think the discussion of figure 7a needs to be fairly addressed. The changes in precipitation rate are quite small over all AR stages, but the greatest difference is in "Before landfall". However, the largest difference is about 0.1mm/hr. Again, the % change is largely irrelevant if the initial magnitude of precipitation rate is very small. The current discussion overemphasizes the impact of marine INP in this assessment.

- What the reviewer described for Figure 7a is accurate and we believe that is what we have in the text written based on the data shown in Figure 7a and Table 1 for the three different stages. About the small precipitation at the stages before AR landfall and post-AR, we clearly noted this, i.e., "Note that precipitation is very small at some point before AR landfall, so the large increases might not mean that much. The total precipitation at the post-AR stage is negligible and the change in domain-mean precipitation from DM15 and DM15+MC18 is also small". For after AR landfall, we clearly stated "There is only a 4% increase in the total precipitation after AR landfall ", and "After AR landfall, precipitation increases significantly. Although the total precipitation is not changed much by the marine INPs, the marine INPs produce a spillover effect featuring reduced precipitation on the windward slope of the mountains but increase precipitation over the lee side (Fig. 8b and Fig. 9e)." We believe those sentences clearly describe the impact of marine INP, which varies at different stages as we noted, and they are in the right place and narrative flow.

- In addition, we do not agree with the reviewer's opinion "the % change is largely irrelevant if the initial magnitude of precipitation rate is very small". We already clearly stated that the small magnitude of precipitation rate results in some large values in the % change (Line 439-440). But the % change does help us quantify how the light precipitation can be influenced by marine INP. Let's explain by an example, the rain rate is 0.05 mm/h averaged over all domain grids (about 400*500 grids) over one hour 00-01 Feb 6 and increases to 0.1 mm/h by marine INP. This means that the accumulated rain over the domain increases from 10 cm to 20 cm within that hour, which is a significant change for light rain amount and in fact it means more to agriculture than heavy rain. Providing the information of the rain rate in the absolute value and % change is useful for readers to judge the significance.

6. Figure 8: Again, please show a difference plot (perhaps as a 3rd column) so we can better visualize the spillover effect. This is very important to emphasize, particularly since the changes in total precip and precip rate are quite small.

- We have considered your suggestion carefully but we think the spillover effect is clearly represented in several places so decided that the additional panel is not needed. First, here in Figure 8, the spillover effect is highlighted with the red rectangle box and it is very clear (drastically different colors). Second, the spillover effect is shown again with Figure 9e which is crystal clear. Third, the newly added difference plot in Fig. 4b (right) also very clearly shows it. Adding another column to Figure 8 is not necessary given there are three figures showing the effect very clearly, and it would just make the spatial map panels too small to be seen clearly.

7. Line 564: Please refrain from using "invigorated" to refer to an increase in condensate. This term is typically referring to dynamics and strength of a system, particularly vertical velocity.

- We actually defined "invigorated cloud cell" as the increase in both LWC and IWC in the paper. To avoid confusion with dynamical changes, we have removed the term in the revised manuscript.

---

## Author Response (AR3)

We thank the editor and the reviewer for the additional comments. We have addressed the comments and please find our responses below.

The editor's comments of:
"Impacts of ice-nucleating particles from marine aerosols on mixed-phase orographic clouds during 2015 ACAPEX field campaign"

The reviewer is satisfied with the revisions that address the comments. Some additional minor figure changes are recommended, e.g., changing Figure 8 to include the difference plot. The same might be done for Figure 7.

We have followed the reviewer's suggestion and added the absolute difference in precipitation volume (precipitation rate multiplies surface area) in Figure 7a (black dotted line) and the absolute difference in total precipitation to Figure 8 (right column). Figure captions have been changed correspondingly. The related text changes for Figure 7a are shown in Lines 439-446. No text change is needed for Figure 8.

I have a minor comment on the introduction of previous work on marine INPs. (Line 89-90) "A few previous studies investigated the impacts of marine INPs on precipitation and radiation with global climate models (Hoose et al., 2010; Burrows et al., 2013; Yun and Penner, 2013).."
Some more recent work on marine INPs effects needs to be included such as Zhao et al. (2021), among others:
Zhao, X., Liu, X., Burrows, S. M., and Shi, Y.: Effects of marine organic aerosols as sources of immersion-mode ice-nucleating particles on high-latitude mixed-phase clouds, Atmos. Chem. Phys., 21, 2305–2327, https://doi.org/10.5194/acp-21-2305-2021, 2021.

We have included three new publications on the effects of marine INP, including Zhao et al. (2021), Burrows et al. (2022), and Shi et al. (2022) (Line 91).

Refs added:

Burrows, S. M., Easter, R. C., Liu, X., Ma, P. L., Wang, H., Elliott, S. M., Singh, B., Zhang, K., and Rasch, P. J.: OCEANFILMS (Organic Compounds from Ecosystems to Aerosols: Natural Films and Interfaces via Langmuir Molecular Surfactants) sea spray organic aerosol emissions – implementation in a global climate model and impacts on clouds, Atmos. Chem. Phys., 22, 5223-5251, 10.5194/acp-22-5223-2022, 2022.

Shi, Y., Liu, X., Wu, M., Zhao, X., Ke, Z., and Brown, H.: Relative importance of high-latitude local and long-range-transported dust for Arctic ice-nucleating particles and impacts on Arctic mixed-phase clouds, Atmos. Chem. Phys., 22, 2909-2935, 10.5194/acp-22-2909-2022, 2022.

Zhao, X., Liu, X., Burrows, S. M., and Shi, Y.: Effects of marine organic aerosols as sources of immersion-mode ice-nucleating particles on high-latitude mixed-phase clouds, Atmos. Chem. Phys., 21, 2305-2327, 10.5194/acp-21-2305-2021, 2021.

Third Review of:
"Impacts of ice-nucleating particles from marine aerosols on mixed-phase orographic clouds during 2015 ACAPEX field campaign"

Thank you for addressing my concerns and suggestions in your responses. I am mostly satisfied with the responses and the changes to the manuscript. Some of your additional statements help provide fairer assessments when comparing the model to observations, and they more fairly show how new parameterizations can improve certain fields while making others less comparable to the obs.

I very much like the addition of the difference plots in figure 4b that clearly demonstrate the spillover effect. It is much easier to see this effect via the difference plots than just "eyeballing" the differences in figure 4a. It would be helpful to do the same for figure 8, but I am not formally requesting this. I just think it would improve the paper and make the differences more easily visible to the reader.

I still disagree with the overemphasis in the % change in very small precipitation rates in figure 7a, but readers can see this for themselves. A plot of absolute increase in precipitation volume over time might provide a clearer impact, but I am not formally requesting this either.

We have followed the reviewer's suggestion and added the absolute difference in precipitation volume (precipitation rate multiplies surface area) in Figure 7a (black dotted line) and the absolute difference in total precipitation to Figure 8 (right column). Figure captions have been changed correspondingly. The related text changes for Figure 7a are shown in Lines 439-446. No text change is needed for Figure 8.